# AceSearcher: Bootstrapping Reasoning and Search for LLMs via Reinforced Self-Play

**Ran Xu**[1]    **Yuchen Zhuang**[2]    **Zihan Dong**[3]    **Jonathan Wang**[1]    **Yue Yu**[2]

**Joyce C. Ho**[1]    **Linjun Zhang**[3]    **Haoyu Wang**[4]    **Wenqi Shi**[5]    **Carl Yang**[1]

[1]Emory University    [2]Georgia Institute of Technology    [3]Rutgers University
[4]SUNY Albany    [5]UT Southwestern Medical Center
🤗 **Dataset/Model:** https://huggingface.co/AceSearcher
⭕ **Code:** https://github.com/ritaranx/AceSearcher/

## Abstract

Search-augmented LLMs often struggle with complex reasoning tasks due to ineffective multi-hop retrieval and limited reasoning ability. We propose AceSearcher, a cooperative self-play framework that trains a single large language model (LLM) to alternate between two roles: a decomposer that breaks down complex queries and a solver that integrates retrieved contexts for answer generation. AceSearcher couples supervised fine-tuning on a diverse mixture of search, reasoning, and decomposition tasks with reinforcement fine-tuning optimized for final answer accuracy, eliminating the need for intermediate annotations. Extensive experiments on three reasoning-intensive tasks across 10 datasets show that AceSearcher outperforms state-of-the-art baselines, achieving an average exact match improvement of 7.6%. Remarkably, on document-level finance reasoning tasks, AceSearcher-32B matches the performance of the giant DeepSeek-V3 model using less than 5% of its parameters. Even at smaller scales (1.5B and 8B), AceSearcher often surpasses existing search-augmented LLMs with up to $9\times$ more parameters, highlighting its exceptional efficiency and effectiveness in tackling complex reasoning tasks.

## 1 Introduction

Large language models (LLMs) have demonstrated remarkable performance in areas such as natural language generation [65, 80, 17] and complex reasoning [26, 20]. However, they often fall short when handling long-tailed or dynamically evolving knowledge [48]. To address these limitations, a growing body of work has explored augmenting LLMs with external search tools that retrieve relevant information at inference time. Search-augmented LLMs not only improve factual accuracy [2, 58], but also facilitate efficient adaptation to new tasks and domains without costly parameter updates [78].

Despite notable advances in retrieval-augmented generation (RAG) [2, 39, 45, 86, 50], most existing approaches are restricted to relatively simple questions [33, 48] solvable through single-turn retrieval. However, real-world applications often demand more complex reasoning, requiring (*i*) multi-hop retrieval to gather relevant evidence from large corpora due to the low recall of direct single-step retrieval [83], and (*ii*) reasoning capability to integrate multiple pieces of information beyond span extraction for response generation [9]. To address these challenges, prior works propose multi-step search via iterative prompting [68, 31, 36, 42, 88], often relying on powerful, closed-source LLMs with strong reasoning abilities. Alternatively, tree-search algorithms have been explored to improve

retrieval and reasoning at inference time [27, 72, 77], but at the expense of increased latency. Recent efforts employing reinforcement learning (RL) frameworks allow LLMs to interact with search engines [30, 92, 62, 4, 63]. While promising, these methods are often memory-intensive and thus less practical for deployment in resource-constrained environments. Additionally, their exclusive reliance on QA datasets for supervision limits the broader potential of LLMs to integrate search with complex, multi-step reasoning across a wider range of tasks.

Motivated by these challenges, we aim to develop an efficient, data-centric training recipe to enhance the capabilities of LLMs for reasoning-intensive search scenarios. Inspired by human problem-solving strategies – where complex tasks are decomposed into simpler subproblems [93, 31, 59], we propose AceSearcher that trains LLMs to act as two roles: *decomposer* and *solver*. The *decomposer* breaks down the original question into subquestions to guide retrieval, while the *solver* generates intermediate and final answers by integrating subquestions, their answers, and context.

We then introduce a two-stage fine-tuning framework to train both the *decomposer* and *solver* modules. In the first stage, we perform supervised fine-tuning (SFT) by extending existing open-domain QA datasets with open-source reasoning data. This covers task decomposition and problem-solving in both text and code. This simultaneously boosts the model's ability to extract relevant information from context as well as strengthens its general reasoning capabilities. In the second stage, we apply reinforcement fine-tuning on targeted reasoning and QA tasks, using rewards derived solely from final outputs. To overcome the lack of intermediate annotations, we hypothesize that *better decompositions lead to more accurate answers*. The *solver* is reinforced to produce correct answers based on decompositions and context, while the *decomposer* is optimized to maximize the solver's accuracy. This framework promotes joint structured reasoning across both roles with one unified model, while eliminating dependence on supervision from proprietary frontier models. Notably, AceSearcher achieves strong performance using iterative preference optimization, without relying on memory-intensive online RL training or costly inference-time scaling.

Our contributions can be summarized as follows:

- We introduce AceSearcher, a cooperative self-play framework designed to jointly enhance LLM's capabilities in both search and reasoning. By introducing two roles, namely the *decomposer* and *solver*, AceSearcher equips a single LLM with joint skills of task decomposition and task solving, providing an efficient and flexible solution for complex reasoning in search-augmented settings.

- We propose a two-stage fine-tuning framework that first applies SFT on a mixture of retrieval, reasoning, and decomposition datasets, followed by reinforcement fine-tuning using rewards solely from the final answer to train the *decomposer* and *solver* without intermediate supervision. This approach can be readily applied to LLMs with varying sizes (1.5B - 32B as shown in our study) to enhance the multi-step reasoning ability of search-augmented LLMs.

- We conduct extensive evaluations of AceSearcher covering *three* tasks across *ten* public datasets. Compared to strong baselines, including recent reasoning models and RL-enhanced search LLMs, AceSearcher demonstrates strong empirical performance with 7.6% gain on average. Moreover, AceSearcher demonstrates high parameter efficiency: the 1.5B variant matches the performance of models 10× larger on QA tasks, highlighting its suitability for low-resource settings.

## 2 Related Works

**Reasoning-intensive Search/Retrieval.** Standard RAG pipelines often consider single-step retrieval only and cannot handle complex questions well [34, 58, 18]. To incorporate reasoning into RAG pipelines, earlier research [68, 70, 71, 36, 88] design multi-turn prompting techniques for complex QA. Besides, several works [2, 75] leverage SFT on high-quality chain-of-thoughts to improve the reasoning skills of LLMs, but without explicit task decomposition. Additionally, [27, 77] design reward-guided search during inference time, [22, 79] trains the query refinement model based on the feedback of generator LLMs, and [6, 37] leverage multi-agent fine-tuning to further enhance reasoning performance, but at the cost of serving multiple LLMs in deployment.

**Self-play Finetuning for LLMs.** Self-play [60] is an effective technique that enables LLMs to learn through self-interaction, promoting diverse experience trajectories and prompt coverage. Recent studies have applied self-play to alignment [8, 76, 84], instruction following [13], theorem proving [15], and reasoning [10, 89]. Unlike these works, we consider *collaborative self-play* for complex problem solving, and tailor LLM self-play frameworks specifically for reasoning-intensive RAG applications.

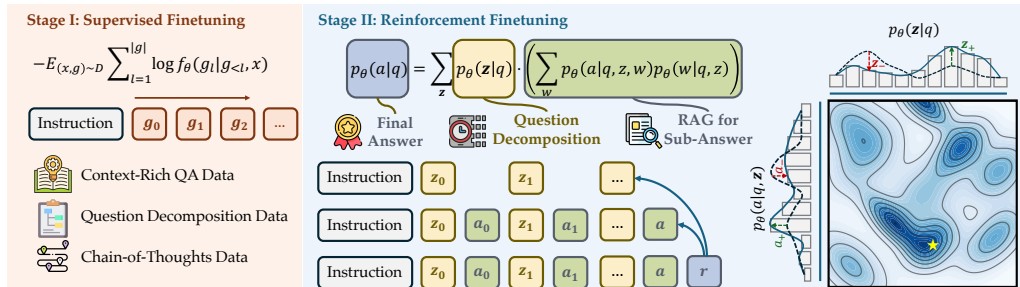

Figure 1: Overview of AceSearcher. AceSearcher contains a two-stage training process to teach LLM for joint precise question decomposition and answer generation with cooperative self-play.

**RL for Search-augmented LLMs.** Very recently (concurrent to us), multiple studies [4, 62, 30, 92, 29] attempted to leverage the RL pipeline for RAG by viewing the search as an external tool and using open-domain QA datasets (e.g. NQ [33], HotpotQA [83]) to create verification rewards. In contrast to these approaches, we propose a data-centric pipeline that enhances LLM retrieval and reasoning capabilities through a unified self-play fine-tuning framework. Our method demonstrates strong generalization across a broad range of reasoning-intensive RAG tasks beyond multi-hop QA.

# 3 Overview of AceSearcher

In this section, we first define the problem setup and present an overview of AceSearcher. Then, we introduce the training and inference pipeline for AceSearcher.

## 3.1 Problem Formulation

In our setting, let $\mathcal{Q}$ denote the space of questions and $\mathcal{A}$ the space of all possible answers. Given a question $q \in \mathcal{Q}$ and corpus $\mathcal{C}$ (e.g. Wikipedia) that provides background knowledge, the retriever (often embedding models) first find a small set of relevant passages $\mathcal{D} = \{d_1, ...d_k\}$, then the LLM $f_\theta$ generates the output $a' \in \mathcal{A}$ conditioned on both $q$ and $\mathcal{D}$ as $a' \sim p_\theta(\cdot \mid q, \mathcal{D})$. Note that $a'$ can be a short- or long-form response, depending on the type of task. In reasoning-intensive scenarios, the question $q$ may require multi-step reasoning beyond simple retrieval to produce accurate answers.

## 3.2 AceSearcher: A Cooperative Self-Play Framework

Our AceSearcher model, shown in Figure 1, tightly couples reasoning and search by enabling a single LLM to act as two roles (controlled by different input and prompt templates):

$\diamond$ **A decomposer** $\rho$ that converts the original question $q$ into a sequence of subquestion templates[1] $z = (z_1, z_2, \ldots, z_n)$, where the number of subquestions is $n$ and $z_i$ may depend on answers to earlier subquestions. These templates are sampled from $z \sim p_\theta(\cdot \mid q)$.

$\diamond$ **A solver** $\pi$ that generates intermediate answers $w = (w_1, w_2, \ldots, w_n)$ and final answer $a'$ in a stepwise manner: For each subquestion $z_i \in z$, the solver produces the *intermediate answer* as $w_i \sim p_\theta(\cdot \mid z_i, w_{<i}, \mathcal{D}_i)$, where $w_{<i}$ denotes the answers to previous subquestions, $\mathcal{D}_i = \{d_{i,1}, \ldots, d_{i,k}\}$ is the set of retrieved passages for $z_i$. After solving subquestions, the solver predicts the *final answer* $a' \sim p_\theta(\cdot \mid q, z, w, \mathcal{D})$ based on the original question, intermediate answers and context passages.

**Joint Learning Objective.** Given the question $q$, we train $\theta$ to maximize the probability of the LLM for generating the final answer $a$. In our framework, the learning objective can be written as

$$p_\theta(a \mid q) = \sum_z p_\theta(z \mid q) \left( \sum_w p_\theta(a \mid q, z, w) \, p_\theta(w \mid q, z) \right) \quad (3.1)$$

*In practice*, marginalizing over all possible decompositions $z$ and intermediate answers $w$ is intractable. To approximate it during training, we sample a small set of candidate $(z, w)$ paths and

---

[1]We refer to them as "templates" since some subquestions are determined by answers to previous ones. An example template [19] is: "Q1: *What items did Aristotle use?*; Q2: *Is laptop in #1?*". In practice, the template $z$ is a text snippet with a fixed format and will be parsed to multiple subquestions, detailed in Appendix F.

identify the most promising ones to encouraging the decomposer to help the solver generate the correct answer. At *inference time*, given a question $q$, the decomposer $\rho$ generates a subquestion sequence $z$, and the solver $\pi$ reasons over the intermediate answer to derive the final answer $a'$.

## 4 Two-Stage Finetuning for AceSearcher

To enable the LLM to perform both roles effectively, we first apply SFT on publicly available datasets to establish its foundational capabilities. Subsequently, we perform reinforcement fine-tuning to further improve LLM's capabilities, using only final answers as supervision.

### 4.1 Stage I: Supervised Finetuning (SFT)

Although recent studies have introduced data mixing strategies for search-augmented LLMs [39, 45, 86, 37], they focus on enhancing the LLM's ability to *extract answers from provided contexts*. In contrast, our setting presents a greater challenge – requiring the LLM to automatically decompose and solve complex questions across a diverse range of tasks that requires reasoning. Towards this goal, we extend the SFT data mixture $\mathcal{D}_{\text{sft}}$ for the following tasks:

- **Context-rich QA Data.** We follow [45, 39, 86, 37] to leverage multiple QA datasets to enhance the LLM's capability of using context for generation. Specifically, we consider the following datasets: NQ [33], SQuAD [56], DROP [16], NarrativeQA [32], Quoref [12], ROPES [38], FEVER [66], TAT-QA [94], which contains a question, context passages, and an answer.

- **Question Decomposition Data.** To improve the LLM's ability to decompose complex questions into simpler subproblems, we incorporate GSM8K [11], ConvFinQA [9], and StrategyQA [19]. These datasets require generating a sequence of subquestions for solving the original problem.

- **Chain-of-thought Data.** To enhance multi-step reasoning, we leverage chain-of-thought datasets including GSM8K [11], TabMWP [46], and IfQA [85]. Inspired by studies showing that combining Chain-of-Thought (CoT) [74] and Program-of-Thought (PoT) [5] rationales can boost reasoning capabilities, we incorporate MathInstruct [87], which contains CoT and PoT style prompts.

Detailed descriptions of datasets, prompt formats, and the number of training examples are provided in Appendix E, F. In total, we curate 180K training examples in the SFT stage. The LLM is fine-tuned using the standard next-token prediction objective.

### 4.2 Stage II: Preference-based Reinforcement Finetuning (RFT)

While SFT equips the LLM with basic capabilities for question decomposition and answer generation, it relies on richly annotated prompts with intermediate question decomposition and chain-of-thought annotations – resources that are limited in practice. To overcome this scarcity, we further fine-tune the LLM on prompts $\mathcal{D} = \{(q, a)\}$ covering RAG and context-reasoning scenarios that contain only the *final answer* $a^*$ given the *question* $q$. We frame this setting as an *interactive environment*, where the LLM learn to actively decompose the question and generate intermediate reasoning steps with external context. This motivates the use of reinforcement learning to optimize the reasoning trajectory in the absence of explicit intermediate supervision.

◇ **Environment for RAG.** We collect labeled pairs from multi-hop QA and fact verification datasets, including HotpotQA [83], 2WikiMHQA [21] and HOVER [28], which require the usage of retrieval to generate accurate answers. To formulate the RAG framework as an environment, the query decomposer $\rho$ first generates a sequence of candidate sub-questions $z = (z_1, \ldots, z_n) \in \mathcal{Q}^n$. For each sub-question $q_i$, $k$ relevant documents are retrieved, denoted by $\mathcal{D}_i$. The solver $p_\theta$ then produces intermediate solutions by generating $w_i \sim p_\theta(\cdot \mid z_i, w_{<i}, \mathcal{D}_i)$, conditioned on the current sub-question, previously generated answers, and retrieved context. Finally, the solver predicts the final answer as $a' \sim p_\theta(\cdot \mid \bigcup_{i=1}^{n} z_i, \bigcup_{i=1}^{n} w_i, \bigcup_{i=1}^{n} \mathcal{D}_i)$.

◇ **Environment for Context-Rich Reasoning.** Beyond RAG-specific tasks, we also focus on improving the LLM's reasoning abilities. To this end, we incorporate three datasets from the SFT stage, including GSM8K [11], TabMWP [46], and ConvFinQA [9], which involve reasoning over contexts $\mathcal{C}$ such as tables, passages, or problem conditions. Under this setting, $\rho$ is used to generate subquestions $z = (z_1, \ldots, z_n) \in \mathcal{Q}^n$, and the solver $p_\theta$ produces intermediate solutions by generating

$w_i \sim p_\theta \left( \cdot \mid z_i, w_{<i}, \mathcal{C} \right)$, conditioned on the current subquestion, previous answers, and contexts. Finally, the solver predicts the final answer as $a' \sim p_\theta (\cdot \mid \bigcup_{i=1}^n z_i, \bigcup_{i=1}^n w_i, \mathcal{C})$.

$\diamond$ **Reward Design.** For both scenarios, the complete trajectory $(q, z_1, w_1, \ldots, z_n, w_n, a)$ is evaluated using a reward signal derived from the final answer. Specifically, the reward function is defined as:

$$r(q, a', a) = \text{EM}(a', a) \times \mathbb{I}(f(q, a') = 1), \tag{4.1}$$

where EM denotes the exact match between the model-generated $a'$ and ground-truth answer $a$. The function $f(q, a')$ represents a format-based binary reward, verifying whether the model generates sub-questions, intermediate answers, and reasoning steps in the correct structure.

**Optimization $\pi_\theta$ and $\rho_\theta$.** During the RL phase, we use the reward function defined above as the feedback to update both $\pi_\theta$ and $\rho_\theta$. Denote $u_\theta(a, z, w|q) = p_\theta(z|q)p_\theta(w, a|q, z)$. Following existing works [52], the overall optimization objective is formulated as

$$\max_\theta \mathbb{E}_q \Big[ \mathbb{E}_{z \sim \rho_\theta, (w, a') \sim \pi_\theta} \left[ r(q, a', a) \right] - \beta \mathbb{D}_{\text{KL}} \left[ u_\theta(a', z, w \mid q) \| u_{\text{ref}}(a', z, w \mid q) \right] \Big], \tag{4.2}$$

where $\beta$ is the parameter for controlling deviation from the reference policy. We further decompose the KL divergence between $u_\theta$ and $u_{\text{ref}}$ as

$$\mathbb{D}_{\text{KL}}(u_\theta \| u_{\text{ref}}) = \sum_{a', z, w} u_\theta(a', z, w|q) \left[ \log p_\theta(z) + \log p_\theta(w, a'|z, q) - \log p_{\text{ref}}(z) - \log p_{\text{ref}}(w, a'|z, q) \right]$$

$$= \underbrace{\sum_z p_\theta(z|q) \left[ \log \frac{p_\theta(z|q)}{p_{\text{ref}}(z|q)} \right]}_{\mathbb{D}_{\text{KL}}(\rho_\theta \| \rho_{\text{ref}})} + \underbrace{\sum_z p_\theta(z|q) \sum_{w, a'} p_\theta(w, a'|z, q) \left[ \log \frac{p_\theta(w, a'|z, q)}{p_{\text{ref}}(w, a'|z, q)} \right]}_{\mathbb{D}_{\text{KL}}(\pi_\theta \| \pi_{\text{ref}})}.$$

Then, the optimization objective can be rewritten as

$$\max_\theta \mathcal{J}_\theta = \mathbb{E}_q \left[ \mathbb{E}_{z \sim \rho_\theta, (w, a') \sim \pi_\theta} [r(q, a', a)] - \beta \, \mathbb{D}_{\text{KL}}(\rho_\theta \| \rho_{\text{ref}}) - \beta \, \mathbb{E}_{z \sim \rho_\theta} [\mathbb{D}_{\text{KL}}(\pi_\theta \| \pi_{\text{ref}})] \right]. \tag{4.3}$$

The above optimization problem have the closed-form solution (details in Appendix B) [55] as

$$p^*(z \mid q) \propto p_{\text{ref}}(z \mid q) \, \mathbb{E}_{(w, a') \sim p_{\text{ref}}(\cdot | q, z)} \left[ \exp \left( \frac{1}{\beta} r(q, a', a) \right) \right],$$

$$p^*(w, a' \mid q, z) \propto p_{\text{ref}}(w, a' \mid q, z) \exp \left( \frac{1}{\beta} r(q, a', a) \right).$$

**What does the form of $\pi^*$ and $\rho^*$ imply?** The closed-form policies $\rho^*$ (i.e. $p^*(z \mid q)$) and $\pi^*$ (i.e. $p^*(w, a' \mid q, z)$) align with our intuitions: an effective decomposition policy $\rho$ promotes *higher overall expected reward* by enabling better intermediate reasoning steps, while an improved solver $\pi$ directly enhances the reward, regardless of the quality of the decomposition.

**Practical Implementation for Optimization.** In practice, direct optimization under sparse reward signals from single-trajectory rollouts is often ineffective due to high variance and limited feedback. We employ a rollout strategy to address this challenge and enrich the learning signal. For each question $q$, we first generate $m$ candidate decompositions by sampling from the decomposer policy, i.e., $z^{(i)} \sim \rho_\theta(\cdot \mid q)$ for $i = 1, \ldots, m$. Then, for each decomposition $z^{(i)}$, we subsequently sample $m'$ candidate solutions by drawing from the solver policy as $a_j \sim \pi_\theta(\cdot \mid q, z^{(i)})$ for $j = 1, \ldots, m'$. To construct preference datasets for RFT, we first identify the best and worst decompositions for each question $q$ based on the expected reward over their corresponding solutions as $\bar{r}(q, z^{(i)}) = \mathbb{E}_{(w, a') \sim \pi(\cdot | q, z^{(i)})} r(q, a', a)$. This results in the following preference pair dataset:

$$\mathcal{D}_{\text{decompose}} = \{(q, z^{(i+)}, z^{(i-)}) | (q, a) \in \mathcal{D}\}, \text{ where } \begin{cases} z^{(i+)} = z^{(j)}, \ j = \arg\max_i \bar{r}(q, z^{(i)}), \\ z^{(i-)} = z^{(j')}, \ j' = \arg\min_i \bar{r}(q, z^{(i)}). \end{cases} \tag{4.4}$$

Constructing preference pairs to optimize the answer generation policy $\pi$ (with fixed subquestions $z$) is more challenging due to the presence of multiple intermediate answers along the reasoning trajectory. Denote the trajectory $\mathcal{T}^{(i)} = (q, z_1, w_1^{(i)}, \ldots, z_n, w_n^{(i)}, a'^{(i)})$ with $a'^{(i)}$ being the final prediction, we create preference pairs for intermediate $\mathcal{D}_{\text{subq}}$ and final question answering $\mathcal{D}_{\text{final}}$ as

$$\mathcal{D}_{\text{subq}} = \cup_{i=1}^n \left\{ (z_i, w_i^+, w_i^-) \mid w_i^+ \neq w_i^-, (q, a) \in \mathcal{D}, (z_i, w_i^+) \in \mathcal{T}^+, (z_i, w_i^-) \in \mathcal{T}^- \right\},$$

$$\mathcal{D}_{\text{final}} = \left\{ \left( [q, z_1, w_1^+, \ldots, z_n, w_n^+], a'^+, a'^- \right) \mid (q, a) \in \mathcal{D} \right\}.$$

where the best and worst trajectories are selected as:

$$\mathcal{T}^+ = (q, z_1, w_1^+, \ldots, z_n, w_n^+, a'^+), \quad \text{where } a'^+ = \arg\max_i r(q, a'^{(i)}, a),$$

$$\mathcal{T}^- = (q, z_1, w_1^-, \ldots, z_n, w_n^-, a'^-), \quad \text{where } a'^- = \arg\min_i r(q, a'^{(i)}, a).$$

To jointly optimize both the decomposer $\rho$ and the solver $\pi$, we construct a unified preference dataset by combining three sources of pairs: $\mathcal{D}_{\text{pref}} = \mathcal{D}_{\text{decompose}} \cup \mathcal{D}_{\text{subq}} \cup \mathcal{D}_{\text{final}}$. For notational consistency, we represent each example as $(x, g^+, g^-)$, where $x$ is the input, and $g^+, g^-$ are the chosen and rejected responses. Following [55], we optimize the policy with the following preference loss:

$$\mathcal{L}_{\text{DPO}} := -\mathbb{E}_{(x,g^+,g^-)\sim\mathcal{D}_{\text{pref}}} \log \sigma \left( \beta \left[ \log \frac{p_\theta(g^+ \mid x)}{p_{\text{ref}}(g^+ \mid x)} - \log \frac{p_\theta(g^- \mid x)}{p_{\text{ref}}(g^- \mid x)} \right] \right).$$

**Multi-turn DPO for Online Optimization.** Motivated by the benefits of on-policy data sampling in RL, we adopt an iterative DPO framework for improved optimization. Specifically, in the $t$-th iteration, we use the LLM policy model[2] $f_\theta^{(t)}$ to act as $\pi_\theta$ and $\rho_\theta$ to sample preference pairs to create the dataset $\mathcal{D}_{\text{pref}}^{(t)}$. Then, we use $\mathcal{D}_{\text{pref}}^{(t)}$ to update the policy model for the next iteration $f_\theta^{(t+1)}$ as

$$\mathcal{L}_{\text{mDPO}} := -\mathbb{E}_{(x,g^+,g^-)\sim\mathcal{D}_{\text{pref}}^{(t)}} \log \sigma \left( \beta \left[ \log \frac{p_\theta^{(t+1)}(g^+ \mid x)}{p_\theta^{(t)}(g^+ \mid x)} - \log \frac{p_\theta^{(t+1)}(g^- \mid x)}{p_\theta^{(t)}(g^- \mid x)} \right] \right). \quad (4.5)$$

**Remark.** To balance *effectiveness* and *efficiency* in practice, we adopt the following strategy: the model $\pi_\theta$ directly generates answers for intermediate questions, while producing a full rationale only for the final answer. To prevent overly long input contexts during final answer generation, we set the total number of documents to $N$ ($N = 15$ in this study), and allocate up to $\lfloor N/n \rfloor$ top-ranked documents for each of $n$ subquestions produced by $\pi_\theta$. We *discard* preference pairs if the reward for the best and the worst response is the same.

**Theorem 4.1** (Informal). Under regularity conditions, with high probability, the minimizer of the loss (Eq. (4.5)) at step $t$ is close to the minimizer of the loss (Eq. (4.2)). Furthermore, as $t$ increases, the minimizer converges to the true parameter $\theta^*$.

The proof for the theorem is deferred to Appendix C due to the space limit. This theorem implies that our optimization algorithm is equivalent to maximizing the reward in Eq. (4.2). Furthermore, it guarantees convergence of our algorithm, which we also empirically validate in Section 5.4.

## 5 Experiments

In this section, we conduct experiments on various tasks to verify the effectiveness of AceSearcher.

### 5.1 Experiment Setups

**Tasks and Dataset Information.** We consider the following 3 types of tasks: (i) *Multi-hop QA*, which includes 2WikiMHQA [21], HotpotQA [83], Bamboogle [53] and MusiQue [67]. (ii) *Multi-hop Fact Verification*, namely HOVER [28] and Exfever [47]. (iii) *Document-level Reasoning*, where we use the DocMath-Eval benchmark [91] with several financial reasoning datasets such as TAT-QA [94], FinQA [7], MultiHiertt [90], and TAT-HQA [35]. Note that some of datasets have very long contexts that make retrieval necessary. The detailed information for these datasets is in Appendix D.

**Baselines.** For *Multihop QA and Fact Verification* tasks, we compare against the following categories of baselines: (i) *Instruction-tuned LLMs with Single-turn RAG*: we consider Llama-3.1-it [17], DeepSeek-R1-Distill [20], Qwen-3 [80][3], Llama-4-Maverick [1], GPT-4o [25], and GPT-4.1 [51]. (ii) *Prompt-based Multi-step Retrieval*: we include IRCOT [68], Plan-RAG [70], Search-o1 [36], IterDRAG [88]. (iii) *Finetuned LLMs with Search*: we compare with InstructRAG [75], RAG-Star [27], ReARTeR [64], CORAG [72] and Iter-RetGen [57]. (iv) *LLMs with Search Trained via Reinforcement Learning*: Recent agentic search-augmented works such as Search-R1 [30], R1-Searcher [62], DeepResearcher [92], MMOA-RAG [6], and ReSearch [4] are also included for comprehensive evaluation. For *document-level reasoning*, we follow DocMath-Eval [91] to compare

---

[2]We denote the model after the SFT stage described in Section 4.1 as $f_\theta^{(1)}$.

[3]For Qwen-3, we evaluate both thinking and non-thinking prompting modes and report the better result.

Table 1: Comparison of AceSearcher and baselines on Multi-hop QA and Fact Verification datasets. "–" stands for results that are not publicly available. $^\dagger$: This model often does not follow instructions and generates long answers. $^\ddagger$: Concurrent works (preprint appears online after 2025/03/01).

| Baselines | 2WikiMHQA | | | HotpotQA | | | Bamboogle | | | MusiQue | | | Hover | ExFever | Avg. QA | Avg. All |
|---|---|---|---|---|---|---|---|---|---|---|---|---|---|---|---|---|
| | Acc | EM | F1 | Acc | EM | F1 | Acc | EM | F1 | Acc | EM | F1 | EM | EM | Acc / EM | EM |
| *Base Size: < 10B parameters* | | | | | | | | | | | | | | | | |
| Llama-3.1-it RAG 8B [17] | 43.0 | 16.0 | 26.5 | 46.2 | 24.4 | 34.5 | 24.8 | 5.6 | 15.1 | 19.8 | 7.2 | 12.5 | 66.3 | 45.0 | 33.5 / 13.3 | 27.4 |
| R1-Distill RAG 8B [20] | 50.8 | 30.0 | 42.8 | 45.2 | 25.2 | 36.2 | 39.2 | 25.6 | 37.3 | 21.6 | 12.4 | 21.4 | 63.0 | 48.2 | 39.2 / 23.3 | 34.1 |
| Qwen-3 RAG 8B‡ [80] | 54.2 | 35.4 | 46.4 | 56.0 | 42.0 | 55.1 | 50.4 | 33.6 | 46.1 | 26.2 | 15.2 | 23.8 | 65.7 | 68.8 | 46.7 / 31.6 | 43.5 |
| Plan-RAG 8B [70] | 47.8 | 36.6 | 46.0 | 47.6 | 35.2 | 45.1 | 31.0 | 23.4 | 32.2 | 20.4 | 12.2 | 21.2 | 57.9 | 62.5 | 36.7 / 26.9 | 38.0 |
| Search-R1 7B‡ [30] | — | 38.2 | — | — | 43.3 | — | — | 43.2 | — | — | 19.6 | — | — | — | — / 36.1 | — |
| R1-Searcher 7B‡ [62] | 63.6 | — | — | 65.4 | — | — | 52.8 | — | — | 28.2 | — | — | — | — | 52.5 / — | — |
| DeepResearcher 7B‡ [92] | 66.6 | — | 59.7 | 64.3 | — | 52.8 | 72.8 | — | 71.0 | 29.3 | — | 27.1 | — | — | — / — | — |
| InstructRAG 8B [75] | 58.6 | 43.2 | 49.5 | 54.6 | 44.0 | 54.4 | 35.2 | 24.8 | 35.5 | 21.2 | 14.8 | 22.8 | 65.3 | 58.0 | 42.4 / 31.7 | 41.7 |
| MMOA-RAG 8B [6] | 42.8 | 41.4 | 46.4 | 39.2 | 36.2 | 48.3 | — | — | — | — | — | — | — | — | — | — |
| CORAG 8B (Greedy) [72] | — | 56.5 | 62.3 | — | 50.1 | 63.2 | — | 37.6 | 51.4 | — | 18.6 | 29.3 | — | — | — / 40.7 | — |
| CORAG 8B (Inference Scaling) [72] | — | 72.5 | 77.3 | — | 56.3 | 69.8 | — | 54.4 | 68.3 | — | 30.9 | 42.4 | — | — | — / 53.5 | — |
| AceSearcher 1.5B | 69.8 | 60.6 | 68.5 | 60.6 | 50.4 | 59.8 | 38.4 | 33.6 | 41.7 | 37.2 | 26.8 | 37.0 | 60.7 | 64.2 | 51.5 / 42.9 | 49.4 |
| AceSearcher 8B | 80.6 | 66.0 | 76.7 | 68.2 | 58.8 | 69.2 | 60.8 | 55.2 | 63.5 | 46.8 | 35.4 | 47.7 | 68.3 | 73.8 | 64.1 / 53.9 | 59.6 |
| *Large Size: 10 - 30B parameters* | | | | | | | | | | | | | | | | |
| Qwen-2.5-it RAG 14B [81] | 44.4 | 17.8 | 29.1 | 50.4 | 35.6 | 48.1 | 40.8 | 22.4 | 35.4 | 21.8 | 9.8 | 18.2 | 65.7 | 42.9 | 39.4 / 21.4 | 32.4 |
| R1-Distill RAG 14B† [20] | 31.8 | 8.4 | 11.4 | 45.6 | 11.2 | 14.6 | 34.8 | 8.0 | 12.1 | 20.8 | 1.8 | 5.2 | 68.3 | 62.5 | 33.3 / 7.4 | 26.7 |
| Qwen-3 RAG 14B‡ [80] | 59.2 | 42.0 | 49.1 | 63.8 | 44.6 | 55.1 | 50.4 | 36.8 | 46.7 | 32.4 | 15.0 | 25.6 | 67.5 | 70.5 | 51.5 / 34.6 | 46.1 |
| Plan-RAG 14B [70] | 61.6 | 51.0 | 60.8 | 60.0 | 48.2 | 59.5 | 51.2 | 41.4 | 53.2 | 34.2 | 23.4 | 32.4 | 52.5 | 63.6 | 51.8 / 41.0 | 46.7 |
| Search-R1 14B‡ [30] | — | 47.0 | — | — | 46.8 | — | — | 52.8 | — | — | 24.1 | — | — | — | — / 42.7 | — |
| InstructRAG 14B [75] | 63.2 | 50.4 | 58.1 | 58.2 | 46.6 | 58.0 | 37.6 | 31.2 | 41.4 | 24.6 | 16.2 | 25.5 | 67.5 | 65.3 | 45.9 / 36.1 | 46.2 |
| AceSearcher 14B | 81.2 | 65.6 | 76.6 | 70.8 | 61.2 | 71.8 | 60.0 | 53.6 | 65.6 | 48.6 | 36.2 | 47.4 | 69.3 | 75.0 | 65.2 / 54.2 | 60.1 |
| *XL Size: > 30B parameters* | | | | | | | | | | | | | | | | |
| Qwen-2.5-it RAG 32B | 51.4 | 31.6 | 40.6 | 58.0 | 38.5 | 50.4 | 59.2 | 51.2 | 65.2 | 22.2 | 10.4 | 19.8 | 70.3 | 69.6 | 47.7 / 32.9 | 45.2 |
| R1-Distill RAG 32B | 57.2 | 39.4 | 51.2 | 63.2 | 49.0 | 62.7 | 56.4 | 46.4 | 58.9 | 30.4 | 18.6 | 30.7 | 72.3 | 67.0 | 51.8 / 38.4 | 48.8 |
| Qwen-3 RAG 32B‡ [80] | 61.0 | 39.8 | 51.5 | 65.4 | 49.0 | 62.4 | 56.8 | 40.8 | 53.6 | 32.6 | 19.0 | 30.7 | 70.5 | 65.3 | 54.0 / 37.2 | 47.4 |
| Search-o1 32B [36] | — | 58.0 | 71.4 | — | 45.2 | 57.3 | — | 56.0 | 67.8 | — | 16.6 | 28.2 | 68.3 | 74.4 | — / 44.0 | 53.1 |
| Plan-RAG 32B [70] | 62.0 | 52.4 | 63.8 | 61.8 | 49.2 | 60.7 | 60.0 | 53.6 | 62.5 | 37.2 | 25.4 | 35.4 | 66.7 | 66.4 | 55.3 / 45.2 | 52.3 |
| ReSearch 32B‡ [4] | — | 45.0 | — | — | 46.7 | — | — | 56.8 | — | — | 26.4 | — | — | — | — / 43.7 | — |
| AceSearcher 32B | 79.0 | 65.6 | 75.3 | 73.8 | 60.4 | 72.7 | 61.6 | 57.6 | 66.6 | 52.2 | 40.2 | 50.8 | 67.0 | 73.2 | 66.7 / 56.0 | 60.7 |
| Qwen-2.5-it RAG 72B | 49.2 | 34.6 | 46.9 | 58.6 | 41.6 | 55.1 | 56.8 | 46.4 | 60.7 | 24.0 | 11.2 | 20.9 | 69.3 | 57.2 | 47.2 / 33.5 | 43.4 |
| R1-Distill RAG 70B | 61.0 | 50.4 | 59.8 | 67.6 | 53.0 | 67.3 | 60.8 | 48.8 | 61.1 | 36.6 | 22.8 | 36.1 | 67.7 | 65.2 | 56.5 / 43.8 | 51.3 |
| Llama-4 Maverick RAG 17B*128‡ [1] | 63.0 | 50.6 | 61.2 | 63.6 | 49.4 | 63.8 | 64.8 | 48.8 | 66.3 | 23.8 | 16.0 | 26.2 | 74.0 | 73.9 | 53.8 / 41.2 | 52.1 |
| *Proprietary Retrieval-Augmented LMs (For reference)* | | | | | | | | | | | | | | | | |
| GPT-4o RAG [25] | 57.8 | 45.8 | 57.2 | 64.0 | 47.2 | 63.6 | 35.2 | 27.2 | 37.2 | 29.8 | 17.4 | 30.0 | 61.7 | 64.8 | 46.7 / 34.4 | 44.0 |
| GPT-4.1 RAG‡ [51] | 51.0 | 42.4 | 49.5 | 60.8 | 44.0 | 59.3 | 40.8 | 35.2 | 44.3 | 30.0 | 18.4 | 29.7 | 67.5 | 66.4 | 45.7 / 35.0 | 45.7 |
| IRCOT (zero shot, w/ GPT-4o) [68] | 61.4 | 51.4 | 61.0 | 64.2 | 48.0 | 63.7 | 60.8 | 46.4 | 56.9 | 33.8 | 22.4 | 33.5 | 63.7 | 64.8 | 55.1 / 42.1 | 49.5 |
| IRCOT (few shot, w/ GPT-4o) [68] | 78.0 | 62.2 | 72.9 | 66.4 | 52.8 | 66.0 | 66.4 | 57.6 | 70.2 | 46.2 | 30.4 | 44.9 | 70.2 | 70.5 | 64.3 / 50.8 | 57.3 |
| Iter-RetGen (w/ GPT-4o) [57] | 71.4 | 52.8 | 69.6 | 62.6 | 48.4 | 63.4 | 62.4 | 48.8 | 67.7 | 40.8 | 26.6 | 42.6 | 68.3 | 69.6 | 59.3 / 44.2 | 52.4 |
| RAG-Star (w/ GPT-4o) [27] | 68.0 | 47.0 | 62.8 | 57.0 | 48.0 | 68.6 | — | — | — | 40.0 | 29.0 | 43.5 | — | — | — | — |
| ReARTeR (w/ GPT-4o-mini) [64] | 53.4 | — | — | 50.6 | — | — | 54.4 | — | — | 30.2 | — | — | — | — | — | — |
| IterDRAG (Gemini-1.5, 5M ctx) [88] | 76.9 | 67.0 | 75.2 | 56.4 | 51.7 | 64.4 | 68.8 | 65.6 | 75.6 | 30.5 | 22.5 | 35.0 | — | — | 58.2 / 51.7 | — |

against general instruction-tuned LLMs [51, 25, 65, 40, 41, 17], reasoning LLMs [26, 20, 80], Code LLMs [96, 24], Math LLMs [43, 82] and specialized finance reasoning LLMs [44, 95].

**Implementation Details.** We consider four different backbones for AceSearcher with varying sizes including `Qwen-2.5-Instruct-1.5B/14B/32B` [81] and `Llama-3.1-8B-Instruct` [17]. For AceSearcher-32B, we apply LoRA fine-tuning [23] with $r = 8, \alpha = 16$, while other models use full fine-tuning. All models are trained with a batch size of 64 and maximum token of 2048 for 1 epoch on both SFT and RFT stages, with RFT run for 2 total iterations. For HotpotQA, 2WikiMHQA, MusiQue, we use the corpora provided by their respective sources. For Bamboogle, Hover, ExFever, we use the Wikipedia from Dec. 2018 as the corpus. During inference, we set the temperature $t = 0.0$, the number of retrieved passages to $k = 10$. For QA and fact verification tasks, we adopt E5 [73] as the retriever, while for document-level reasoning, we follow [91] and use OpenAI's `Embedding-3-Large` as the retriever. Detailed implementation settings for AceSearcher and baselines are in Appendix G.

**Evaluation.** For QA, we report Exact Match (EM), Accuracy, and F1 score. For fact verification, we use EM as the metric. For document-level reasoning, we use Accuracy computed via the official evaluation script, and report the better performance between CoT and PoT prompting [54].

## 5.2 Evaluation on QA and Fact Verification

The main results comparing AceSearcher with baseline methods are presented in Table 1. From the results, we have the following key observations: (i) **AceSearcher achieves strong performance over baselines.** Notably, AceSearcher-32B achieves the highest overall score (60.7), outperforming both proprietary and open-source baselines by up to 7.6%. (ii) **Compared to reasoning models, AceSearcher better adapt to RAG tasks.** Qwen-3 and Deepseek-R1-distill are trained with extensive knowledge distillation, we observe that their gains are limited. This suggests that long thinking does not fully address the inherent challenge of multi-hop retrieval, while AceSearcher tackles this more effectively. (iii) **AceSearcher has strong parameter efficiency.** AceSearcher-1.5B matches or exceeds 8B baselines, while AceSearcher-8B outperforms baseline models with 70B parameters.

Table 3: Ablation results on QA and DocMath-Eval using Llama-3.1-8B. We report EM for QA and fact verification due to space constraints. For *w/o ρ* and *w/o π*, we replace the respective components with `Llama-3.1-8B-Instruct`. *w/o Search* excludes CQA, StrategyQA, and IfQA from SFT; *w/o Reasoning* removes GSM8K, TabMWP, ConvFinQA, and MathInstruct. *w/ CQA* follows [45] and finetune solely on context-aware QA tasks.

| Model Name | 2WikiMHQA | HotpotQA | Bamboogle | MusiQue | Hover | ExFever | Avg. | DM$_{SS}$ | DM$_{CS}$ | DM$_{SL}$ | DM$_{CL}$ | Avg. |
|---|---|---|---|---|---|---|---|---|---|---|---|---|
| *Ablation Study for Different Components of AceSearcher* | | | | | | | | | | | | |
| AceSearcher | 66.0 | 58.8 | 55.2 | 35.4 | 68.3 | 73.8 | 59.6 | 83.0 | 80.5 | 48.0 | 32.3 | 59.0 |
| AceSearcher w/o RFT | 61.8 | 53.8 | 52.0 | 34.8 | 64.1 | 71.4 | 56.2 | 71.5 | 73.0 | 49.0 | 26.7 | 52.3 |
| AceSearcher w/o SFT | 40.0 | 37.2 | 38.4 | 18.2 | 74.7 | 76.8 | 47.6 | 71.0 | 51.5 | 46.0 | 31.0 | 48.0 |
| AceSearcher w/o ρ | 57.8 | 52.4 | 53.6 | 32.4 | 65.0 | 70.5 | 55.3 | 81.5 | 78.0 | 45.0 | 29.6 | 56.6 |
| AceSearcher w/o π | 41.4 | 32.0 | 20.8 | 12.2 | 63.7 | 75.0 | 40.9 | 73.5 | 72.0 | 45.0 | 27.7 | 52.4 |
| *Ablation Study for SFT Data Mixture* | | | | | | | | | | | | |
| AceSearcher w/o Search | 52.6 | 53.0 | 51.2 | 23.4 | 56.5 | 58.9 | 49.3 | 79.5 | 83.0 | 42.0 | 28.7 | 56.6 |
| AceSearcher w/o Reasoning | 62.4 | 55.8 | 44.8 | 36.6 | 57.7 | 71.4 | 54.8 | 76.5 | 74.0 | 38.0 | 29.7 | 53.5 |
| AceSearcher w/ CQA [45] | 35.8 | 40.0 | 22.4 | 12.2 | 61.6 | 45.7 | 36.3 | 53.0 | 52.0 | 38.0 | 20.3 | 38.6 |
| *Ablation Study for RL Algorithms* [†] | | | | | | | | | | | | |
| RAFT [14] | 63.6 | 55.6 | 50.4 | 32.8 | 66.7 | 69.6 | 56.5 | 73.0 | 69.5 | 43.0 | 27.3 | 51.2 |
| REST$^{EM}$ [61] | 65.4 | 57.6 | 51.2 | 32.8 | 67.5 | 68.7 | 57.2 | 77.5 | 81.0 | 48.0 | 28.0 | 56.1 |
| Offline DPO [55] | 64.6 | 57.8 | 53.6 | 35.2 | 64.6 | 73.2 | 58.2 | 73.5 | 83.5 | 49.0 | 30.0 | 56.6 |
| (Iterative) SimPO [49] | 67.2 | 57.2 | 46.8 | 34.6 | 69.3 | 70.5 | 57.6 | 75.5 | 78.0 | 44.0 | 32.0 | 55.9 |

## 5.3 Evaluation on Document-level Reasoning

We evaluate AceSearcher on DocMath-Eval (Table 2) against large-scale LLMs, demonstrating notable improvements over similarly-sized baselines, including both reasoning and domain-specific models. For instance, AceSearcher-32B and AceSearcher-8B outperform size-comparable baselines by 6.2% and 9.0%, respectively. Furthermore, AceSearcher achieves performance comparable to significantly larger models: AceSearcher-32B matches the accuracy of DeepSeek-V3 using less than 5% of its parameters, while AceSearcher-14B exceeds baselines up to 72B (5×) in size. These results highlight AceSearcher's strong generalization capabilities beyond factual QA, particularly in complex reasoning scenarios involving long documents and tables.

## 5.4 Additional Studies

**Ablation Study.** Table 3 reports the results of AceSearcher. The top rows show that both SFT and RFT contribute to overall performance gains. Besides, AceSearcher improves both question decomposition (ρ) and answer generation (π), as replacing each component with the frozen Llama-8b-it hurts the performance. This verifies the complementary roles of these two components.

**Ablation Study For SFT Data.** The middle rows in Table 3 show SFT performance under different data compositions. Removing either the Reasoning or Search data leads to performance drops across both knowledge-intensive tasks (QA and Fact Verification) and document-level reasoning, indicating that both components are jointly beneficial for building a capable LLM with search.

**Ablation Study For RFT.** In the bottom rows of Table 3, we compare our reinforcement finetuning algorithm with other alternatives and find AceSearcher achieves the best performance. This highlights the importance of using both positive and negative trajectories, and shows that online methods outperform their offline counterparts. Figure 2 shows results across RFT iterations of

Table 2: Results on DocMath-Eval [91], sorted by average performance. SS, CS, SL and CL stands for SimpShort, CompShort, SimpLong and CompLong, respectively.

| Datasets | DM$_{SS}$ | DM$_{CS}$ | DM$_{SL}$ | DM$_{CL}$ | Avg. |
|---|---|---|---|---|---|
| *Proprietary Models* | | | | | |
| GPT-o3-mini | 86.0 | **87.5** | 59.0 | 35.0 | 63.9 |
| Gemini-1.5-Pro | 85.5 | 80.0 | 58.0 | 40.3 | 63.7 |
| GPT-4.1‡ | 85.5 | 75.0 | 62.0 | 39.3 | 62.6 |
| GPT-4o | 86.0 | 76.5 | **64.0** | 36.7 | 62.4 |
| Claude-3.5-Sonnet | 78.0 | 76.0 | 54.0 | **44.0** | 61.8 |
| *Open-Sourced Models* | | | | | |
| DeepSeek-V3 685B | **89.5** | 86.0 | 53.0 | 42.3 | **66.4** |
| AceSearcher 32B | **89.5** | 84.0 | 53.0 | 43.0 | 66.1 |
| DeepSeek-V2 236B | 87.0 | 75.5 | 61.0 | 43.0 | 64.4 |
| DeepSeek-R1 685B | 89.0 | 83.5 | 53.0 | 38.7 | 64.3 |
| DeepSeek-Coder-V2 236B | 85.0 | 78.0 | 56.0 | 41.0 | 63.1 |
| Mistral-Large 122B | 85.0 | 76.5 | 56.0 | 41.0 | 62.8 |
| AceSearcher 14B | 84.0 | 82.0 | 49.0 | 39.3 | 62.4 |
| AceMath 72B | 77.5 | 77.0 | 59.0 | 39.7 | 60.6 |
| Qwen-2.5-Math 72B | 78.0 | 73.0 | 58.0 | 41.0 | 60.4 |
| DianJin-R1 32B‡ | 76.0 | 77.0 | 46.0 | 42.3 | 59.9 |
| AceSearcher 8B | 83.0 | 80.5 | 48.0 | 32.3 | 59.0 |
| Qwen-2.5-Coder 32B | 81.0 | 79.0 | 57.0 | 30.0 | 58.4 |
| DeepSeek-R1-Distill 70B | 77.5 | 76.0 | 53.0 | 34.7 | 58.0 |
| Qwen-2.5 72B | 81.5 | 81.0 | **64.0** | 24.7 | 57.9 |
| DeepSeek-R1-Distill 32B | 74.0 | 71.0 | 50.0 | 40.3 | 57.6 |
| Llama-3.3 70B | 79.5 | 74.5 | 54.0 | 31.7 | 57.1 |
| Qwen3 32B‡ | 80.0 | 78.0 | 44.0 | 25.3 | 54.5 |
| Qwen3 14B‡ | 75.0 | 78.5 | 41.0 | 26.7 | 53.5 |
| DianJin-R1 7B‡ | 67.0 | 68.5 | 41.0 | 29.3 | 50.4 |
| AceMath 7B | 65.5 | 62.0 | 47.0 | 26.7 | 47.8 |
| AceSearcher 1.5B | 66.5 | 77.5 | 39.0 | 18.0 | 47.6 |
| Qwen3 8B‡ | 76.0 | 76.5 | 32.0 | 11.7 | 46.5 |
| Fin-R1 7B‡ | 66.5 | 51.5 | 40.0 | 21.3 | 42.5 |
| DeepSeek-Coder-V2 16B | 67.5 | 53.5 | 30.0 | 20.3 | 41.6 |
| Llama-3.1 8B | 62.0 | 44.0 | 32.0 | 19.0 | 37.6 |
| Qwen-2.5-Math 7B | 52.0 | 49.0 | 36.0 | 16.7 | 36.0 |

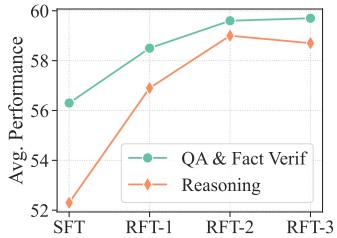

Figure 2: Performance of AceSearcher over different stages.

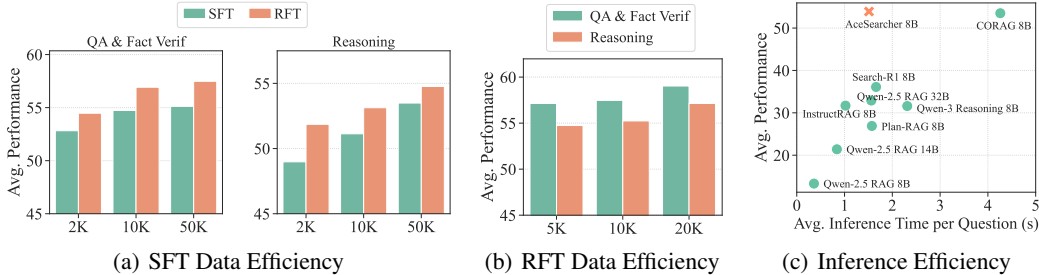

(a) SFT Data Efficiency      (b) RFT Data Efficiency      (c) Inference Efficiency

Figure 3: Efficiency Studies of AceSearcher with `Llama-3.1-8B-Instruct` as the backbone.

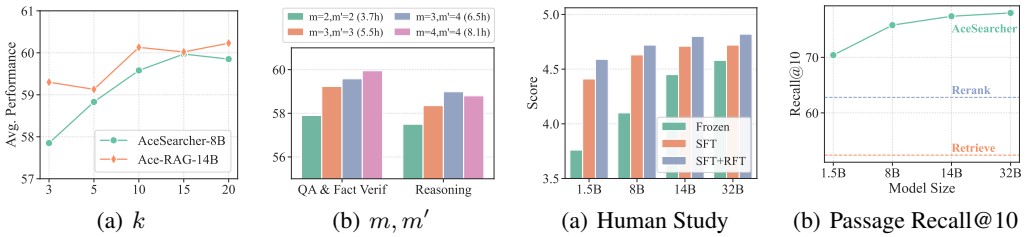

(a) $k$      (b) $m, m'$      (a) Human Study      (b) Passage Recall@10

Figure 4: Parameter Study      Figure 5: Quality Analysis for AceSearcher

AceSearcher-8B. We observe significant gains in the first two iterations, with diminishing returns in the third. We set the number of iterations to 2 to balance between performance and efficiency.

## 5.5 Efficiency Studies

**Data Efficiency.** Figure 3(a) and 3(b) show the accuracy of AceSearcher under varying amounts of data. For SFT, we evaluate the performance with varying SFT subset sizes and its improvement after subsequent RFT. For RFT experiments, we fix the full SFT dataset to isolate the effect of RFT. With just 2K SFT examples (∼1%), AceSearcher matches strong baselines like Search-R1 and Search-O1 (with up to 4× more parameters), and surpasses them after RFT. In the RFT stage, the use of only 5K prompts leads to a 1% gain in QA and fact verification and a 2% gain on document-level reasoning, justifying the data efficiency of AceSearcher with a diverse set of prompts.

**Inference Efficiency.** Figure 3(c) shows the inference time of AceSearcher and baseline models on QA and fact verification tasks. Unless noted, all models are 8B or similar in size. While AceSearcher incurs higher latency than standard RAG due to question decomposition and multi-step reasoning, it achieves substantial performance gains – even outperforming 32B models with comparable inference time. Besides, AceSearcher outperforms reasoning models and inference-time scaling methods costs 1.5× to 2.8× more time. These results justify AceSearcher balances between efficiency and efficacy.

## 5.6 Parameter Studies

We study the effect of varying $k$, $m$, and $m'$ on AceSearcher. As shown in Figure 4(a), performance improves with more retrieved contexts, with gains plateauing at $k = 10$, which we adopt in our experiments. In Figure 4(b) shows that increasing the number of sampled decompositions ($m$) and final answers ($m'$) generally improves performance as it will generate more valid preference pairs, but increases trajectory collection time. The study on the effect of $\beta$ and retrievers is in Appendix H.

## 5.7 Quality Analysis of Question Decomposition Module

As question decomposition is a key component of AceSearcher, we analyze the quality of the generated subquestions. Figure 5(a) shows the average human evaluation scores (on a 1–5 scale) for 40 randomly sampled subquestions per task. We observe that both SFT and RFT significantly enhance subquestion quality across different model sizes. To quantify the impact of decomposition on end-task performance, we evaluate passage-level answer recall on HotpotQA after applying question decomposition. As shown in Figure 5(b), AceSearcher achieves up to a 25% improvement

in recall@10 over standard retrieval and surpasses strong passage reranking model[4]. The details for human studies as well as more cases studies are given in the Appendix I.

## 6 Conclusion

We present AceSearcher, a cooperative self-play framework specifically designed for RAG and document-level reasoning tasks. By training a single LLM to act as both decomposer and solver, AceSearcher addresses complex multi-hop retrieval and reasoning effectively. Our two-stage fine-tuning framework combines SFT on diverse reasoning tasks with preference-based RFT guided by final answer accuracy, achieving strong performance without relying on expensive intermediate supervision. Evaluated on ten benchmarks, AceSearcher outperforms state-of-the-art models by 7.6% on multi-hop QA and fact verification, and matches Deepseek-V3 on document reasoning with under 5% of its parameters. Even with smaller models (1.5B, 8B), AceSearcher delivers competitive or superior performance, offering an efficient and generalizable solution for advanced reasoning under resource constraints.

## Acknowledgment

RX and CY were partially supported by the US National Science Foundation under Award Numbers 2319449, 2312502, and 2442172, as well as the US National Institute of Diabetes and Digestive and Kidney Diseases of the US National Institutes of Health under Award Number K25DK135913. JH was partially supported by the US National Science Foundation (NSF) grant IIS-2145411. WS was partially supported by the Texas Advanced Computing Center (TACC) and the NVIDIA Academic Grant Program. LZ was partially supported by the NSF CAREER DMS-2340241 and AI for Math Fund from Renaissance Philanthropy.

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

# A  Limitations and Impact Statement

**Limitations.** While AceSearcher demonstrates strong empirical performance across a wide range of RAG and document-level reasoning benchmarks, several limitations remain. First, our framework is evaluated primarily on complex QA, fact verification, and document-level reasoning tasks; its applicability to other tasks such as open-ended generation, dialogue, or use of real-time tools remains to be explored, though our scope is comparable (or even broader) compared to concurrent works [62, 30, 92]. Second, AceSearcher relies on a *fixed* retriever during training and inference. Joint optimization of retrieval and reasoning could offer further gains but is left for future work. Third, our decomposition-based pipeline introduces inference overhead, which may limit applicability in latency-sensitive settings. Nonetheless, as shown in Figure 3(c), AceSearcher achieves favorable tradeoffs, and many strong baselines [70, 36, 72] also adopt multi-turn retrieval. Finally, due to resource constraints, we adopt iterative preference optimization (Online DPO) as a practical and efficient alternative to fully online reinforcement learning. While this approach achieves strong results in our setting, exploring more expressive RL formulations may offer further improvements.

**Impact Statement.** This work advances the development of search-augmented LLMs capable of complex reasoning. By enabling smaller open-source LLMs to search and reason more effectively, AceSearcher reduces reliance on proprietary or extremely large models, which may have high computational or financial barriers. This can promote democratization of advanced AI capabilities in low-resource or domain-specific applications, such as finance, scientific discovery, and healthcare.

# B  Derivation Step for Optimal Policy $\pi^*$ and $\rho^*$

We aim to maximize the following objective:

$$\mathcal{J}_\theta = \mathbb{E}_q \Big[ \mathbb{E}_{z \sim \rho_\theta(\cdot|q), \, a \sim \pi_\theta(\cdot|q,z)} [r(q, a', a)] - \beta \, \mathcal{D}_{\mathrm{KL}} \left( \rho_\theta \| \rho_{\mathrm{ref}} \right) - \beta \, \mathbb{E}_{z \sim \rho_\theta(\cdot|q)} \left[ \mathcal{D}_{\mathrm{KL}} \left( \pi_\theta \| \pi_{\mathrm{ref}} \right) \right] \Big].$$

Since $\rho$ and $\pi$ appear in separate terms, we can optimize them independently.

**1. Optimal $\pi$ for each $z$.**  For fixed $z$, consider the Lagrangian

$$\mathcal{L}_z(\pi, \lambda_z) = \sum_{w,a'} \pi(w, a' \,|\, q, z) \, r(q, a', a) - \beta \sum_{w,a'} \pi(w, a' \,|\, q, z) \ln \frac{\pi(w, a' \,|\, q, z)}{\pi_{\mathrm{ref}}(w, a' \,|\, q, z)} \tag{B.1}$$

$$+ \lambda_z \Big( \sum_{w,a'} \pi(w, a' \,|\, q, z) - 1 \Big). \tag{B.2}$$

Taking the functional derivative with respect to $\pi(w, a' \,|\, q, z)$ and setting to zero gives

$$r(q, a', a) - \beta \Big( \ln \pi(w, a' \,|\, q, z) - \ln \pi_{\mathrm{ref}}(w, a' \,|\, q, z) + 1 \Big) + \lambda_z = 0.$$

Rearranging yields

$$\ln \pi(w, a' \,|\, q, z) = \ln \pi_{\mathrm{ref}}(w, a' \,|\, q, z) + \frac{1}{\beta} \, r(q, a', a) + \underbrace{\left( \frac{\lambda_z}{\beta} - 1 \right)}_{\text{constant in } w, a'}.$$

Hence, the optimal policy is:

$$\pi^*(w, a \mid q, z) = \frac{1}{Z_\pi(q, z)} \pi_{\mathrm{ref}}(w, a \mid q, z) \exp \left( \frac{1}{\beta} r(q, a', a) \right)$$

where

$$Z_\pi(q, z) = \sum_{w,a} \pi_{\mathrm{ref}}(w, a \mid q, z) \exp \left( \frac{1}{\beta} r(q, a', a) \right)$$

Now compute $G(\pi^*)$

$$\log \frac{\pi^*(w, a \mid q, z)}{\pi_{\mathrm{ref}}(w, a \mid q, z)} = \frac{1}{\beta} r(q, a', a) - \log Z_\pi(q, z)$$

so

$$r(q, a) - \beta \log \frac{\pi^*(a \mid q, z)}{\pi_{\text{ref}}(a \mid q, z)} = \beta \log Z_\pi(q, z)$$

Therefore,

$$G\left(\pi^*\right) = \mathbb{E}_{w, a \sim \pi^*}\left[\beta \log Z_\pi(q, z)\right] = \beta \log Z_\pi(q, z)$$

**2. Optimal $\rho$.** Substitute $\pi^*$ back into $\mathcal{J}_q$. Denote

$$F[\rho] = \mathbb{E}_{z \sim \rho}\left[\beta \log Z_\pi(q, z) - \beta \log \frac{\rho(z \mid q)}{\rho_{\text{ref}}(z \mid q)}\right] = \beta \mathbb{E}_{z \sim \rho}\left[\log \frac{\rho_{\text{ref}}(z \mid q) Z_\pi(q, z)}{\rho(z \mid q)}\right]$$

together with the constraint $\sum_z \rho(z \mid q) = 1$. Introduce multiplier $\mu$ and form

$$\mathcal{L}[\rho, \mu] = \sum_z \rho(z \mid q) \beta \log Z_\pi(q, z) - \beta \sum_z \rho(z \mid q) \ln \frac{\rho(z \mid q)}{\rho_{\text{ref}}(z \mid q)} + \mu\left(\sum_z \rho(z \mid q) - 1\right).$$

Taking

$$\frac{\partial \mathcal{L}}{\partial \rho(z \mid q)} = \beta \log Z_\pi(q, z) - \beta\left(\ln \rho(z \mid q) - \ln \rho_{\text{ref}}(z \mid q) + 1\right) + \mu = 0.$$

Rearranging:

$$\ln \rho(z \mid q) = \ln \rho_{\text{ref}}(z \mid q) + \log Z_\pi(q, z) + \left(\frac{\mu}{\beta} - 1\right).$$

The optimal policy is:

$$\rho^*(z \mid q) = \frac{1}{Z_\rho(q)} \rho_{\text{ref}}(z \mid q) Z_\pi(q, z)$$

where

$$Z_\rho(q) = \sum_z \rho_{\text{ref}}(z \mid q) Z_\pi(q, z)$$

Combining these two results yields exactly the stated closed-form solutions

$$p^*(z \mid q) \propto p_{\text{ref}}(z \mid q) \mathbb{E}_{(w, a') \sim p_{\text{ref}}(\cdot \mid q, z)}\left[\exp\left(\frac{1}{\beta} r(q, a', a)\right)\right], \qquad \text{(B.3)}$$

$$p^*(w, a' \mid q, z) \propto p_{\text{ref}}(w, a' \mid q, z) \exp\left(\frac{1}{\beta} r(q, a', a)\right). \qquad \text{(B.4)}$$

# C   Omitted Theorems and Proofs

## C.1   Notion

Let $B(r, x)$ represent the $l_2$-ball of radius $r$ centered at $x$. For two positive sequences $\{a_n\}$ and $\{b_n\}$, $a_n \gtrsim b_n$ if $a_n \geq C b_n$. The $l_2$ norm of a vector $x \in \mathbb{R}^d$ is defined as $\|x\|_2 := \left(\sum_{i=1}^d x_i^2\right)^{1/2}$. A sequence of random variables $X_n$ is said to be $o_P(1)$ if $X_n \xrightarrow{P} 0$, that is, $X_n$ converges to 0 in probability as $n \to \infty$. The Kullback–Leibler (KL) divergence from a discrete distribution $p$ to a discrete distribution $q$ (defined over a common support $\mathcal{X}$) is given by $\mathbb{D}_{\text{KL}}\left[p \| q\right] := \sum_{x \in \mathcal{X}} p(x) \log\left(\frac{p(x)}{q(x)}\right)$, under the assumption that whenever $p(x) > 0$, one also has $q(x) > 0$ for all $x \in \mathcal{X}$.

## C.2 Main theorem

Recall the losses (4.2) and (4.5) are defined as follows:

$$\mathbb{E}_q \left[ \mathbb{E}_{z \sim \rho_\theta, (w,a') \sim \pi_\theta} \left[ r(a', q, a) \right] - \beta \mathbb{D}_{\mathrm{KL}} \left[ u_\theta(a', z, w \mid q) \| u_{\mathrm{ref}}(a', z, w \mid q) \right] \right]. \tag{C.1}$$

$$\mathcal{L}_{\mathrm{mDPO}} := -\mathbb{E}_{(x, g^+, g^-) \sim \mathcal{D}_{\mathrm{pref}}^{(t)}} \log \sigma \left( \beta \left[ \log \frac{p_\theta^{(t+1)}(g^+ \mid x)}{p_\theta^{(t)}(g^+ \mid x)} - \log \frac{p_\theta^{(t+1)}(g^- \mid x)}{p_\theta^{(t)}(g^- \mid x)} \right] \right). \tag{C.2}$$

To enable decomposition into a decomposer and a solver, we require the following assumption:

*assumption* C.1 (Conditional Probability decomposition). We assume the following decomposition holds:

$$p_\theta(a \mid q) = \sum_z p_\theta(z \mid q) \left( \sum_w p_\theta(a \mid q, z, w) \, p_\theta(w \mid q, z) \right)$$

We present the informal version of our theorem below. Formal statements are given in Theorems C.2 and C.3.

**Theorem C.1** (Informal). Under regularity conditions, with high probability, the minimizer of the loss (C.2) at step $t$ is close to the minimizer of the loss (C.1). Furthermore, as $t$ increases, the minimizer converges to the true parameter $\theta^*$.

*Remark* C.1. The main theorem can be divided into two components. The first component establishes the equivalence between loss (C.1) and loss (C.2) are equivalent. The second component shows that, once the equivalence is established and the maximizer of loss (C.1) converges, the minimizer of loss (C.2) also converges.

The proof is organized as follows: In Appendices C.3 and C.4, we analyze the convergence properties of the maximizer of the population version loss (C.3) and sample version of loss (C.6) which corresponds exactly to loss (C.1). In Appendix C.5, we demonstrate the equivalence of loss (C.1) to loss (C.2). Finally, in Appendix C.6, building on these results, we prove that the minimizer of loss (C.2) converges as well.

## C.3 Population Version

Based on the loss (C.1), define the population version loss as

$$L(\theta \mid \theta_{t-1}) = \mathbb{E}_{(q,a) \sim p_{\theta^*}(\cdot)} \left[ \mathbb{E}_{z \sim \rho_\theta(\cdot \mid q)} \left[ \mathbb{E}_{(w,a') \sim \pi_\theta(\cdot \mid z, q)} [r(a', q, a)] \right] \right]$$
$$- \beta \mathbb{D}_{\mathrm{KL}} \left( u_\theta(a', z, w \mid q) \| u_{\theta_{t-1}}(a', z, w \mid q) \right). \tag{C.3}$$

Define the operator $M : \Theta \to \Theta$,

$$M(\theta) = \arg \max_{\theta' \in \Theta} L(\theta' \mid \theta),$$

where $\Theta$ represents the parameter space. Notice that it is natural to assume that $\theta^*$ satisfy the self-consistency, i.e. $\theta^* = M(\theta^*)$. So the first assumption will be:

*assumption* C.2 (Self-consistency). $\theta^* = M(\theta^*)$.

*assumption* C.3 ($\lambda$-strong Concavity). There is some $\lambda > 0$ such that

$$L(\theta_1 \mid \theta^*) - L(\theta_2 \mid \theta^*) - \langle \nabla L(\theta_2 \mid \theta^*), \theta_1 - \theta_2 \rangle \le -\frac{\lambda}{2} \|\theta_1 - \theta_2\|_2^2 \quad \text{for all } \theta_1, \theta_2 \in B(r, \theta^*). \tag{C.4}$$

**Definition C.1** (First-order stability). The functions $\{L(\cdot \mid \theta), \theta \in \Theta\}$ satisfy the First-order stability condition over $B(r, \theta^*)$ if

$$\|\nabla L(M(\theta) \mid \theta^*) - \nabla L(M(\theta) \mid \theta)\|_2 \le \mu \|\theta - \theta^*\|_2$$

for all $\theta \in B(r, \theta^*)$.

*assumption* C.4. Assume the functions $\{L(\cdot \mid \theta), \theta \in \Theta\}$ satisfy the First-order stability condition (C.1).

**Proposition C.1** (Population Version). For some radius $r > 0$ and pair $(\mu, \lambda)$ such that $0 \leq \mu < \lambda$, suppose that the Assumption C.1-C.4 hold, then the population operator $M$ is contractive over $B(r, \theta^*)$, in particular with

$$\|M(\theta_{t-1}) - \theta^*\|_2 \leq \frac{\mu}{\lambda}\|\theta_{t-1} - \theta^*\|_2 \quad \text{for all } \theta \in B(r, \theta^*).$$

*Proof.* By the first order optimality condition, we have:

$$\langle \nabla L(\theta^* \mid \theta^*), \theta - \theta^* \rangle \leq 0 \quad \forall \theta$$
$$\Rightarrow \langle \nabla L(\theta^* \mid \theta^*), M(\theta_{t-1}) - \theta^* \rangle \leq 0$$
$$\langle \nabla L(M(\theta_{t-1}) \mid \theta_{t-1}), \theta - M(\theta_{t-1}) \rangle \leq 0 \quad \forall \theta$$
$$\Rightarrow \langle \nabla L(M(\theta_{t-1}) \mid \theta_{t-1}), \theta^* - M(\theta_{t-1}) \rangle \leq 0.$$

Combine the two terms,

$$\langle \nabla L(\theta^* \mid \theta^*) - \nabla L(M(\theta_{t-1}) \mid \theta_{t-1}), M(\theta_{t-1}) - \theta^* \rangle \leq 0.$$

Thus,

$$\langle \nabla L(\theta^* \mid \theta^*) - \nabla L(M(\theta_{t-1}) \mid \theta^*), M(\theta_{t-1}) - \theta^* \rangle \leq$$
$$- \langle \nabla L(M(\theta_{t-1}) \mid \theta^*) - \nabla L(M(\theta_{t-1}) \mid \theta_{t-1}), M(\theta_{t-1}) - \theta^* \rangle.$$

For the right-hand side, by Cauchy-Schwarz inequality,

$$- \langle \nabla L(M(\theta_{t-1}) \mid \theta^*) - \nabla L(M(\theta_{t-1}) \mid \theta_{t-1}), M(\theta_{t-1}) - \theta^* \rangle \leq$$
$$\|\nabla L(M(\theta_{t-1}) \mid \theta^*) - \nabla L(M(\theta_{t-1}) \mid \theta_{t-1})\|_2 \|M(\theta_{t-1}) - \theta^*\|_2.$$

By Assumption C.4,

$$\|\nabla L(M(\theta_{t-1}) \mid \theta^*) - \nabla L(M(\theta_{t-1}) \mid \theta_{t-1})\|_2 \|M(\theta_{t-1}) - \theta^*\|_2 \leq \mu\|M(\theta_{t-1}) - \theta^*\|_2^2.$$

For the left-hand side, by Assumption C.3,

$$\frac{\lambda}{2}\|M(\theta_{t-1}) - \theta^*\|_2^2 \leq L(\theta^* \mid \theta^*) - L(M(\theta_{t-1}) \mid \theta^*) + \langle \nabla L(\theta^* \mid \theta^*), M(\theta_{t-1}) - \theta^* \rangle,$$

$$\frac{\lambda}{2}\|M(\theta_{t-1}) - \theta^*\|_2^2 \leq L(M(\theta_{t-1}) \mid \theta^*) - L(\theta^* \mid \theta^*) + \langle \nabla L(M(\theta_{t-1}) \mid \theta^*), \theta^* - M(\theta_{t-1}) \rangle.$$

Hence,

$$\lambda\|M(\theta_{t-1}) - \theta^*\|_2^2 \leq \langle \nabla L(\theta^* \mid \theta^*) - \nabla L(M(\theta_{t-1}) \mid \theta^*), M(\theta_{t-1}) - \theta^* \rangle. \quad \text{(C.5)}$$

Combining all,

$$\lambda\|M(\theta_{t-1}) - \theta^*\|_2^2 \leq \mu\|M(\theta_{t-1}) - \theta^*\|_2^2.$$

$\square$

*Remark* C.2. This theorem follows the idea in [3]. It suggests that, under a self-play procedure, the algorithm progressively approaches the true underlying distribution. This behavior is characterized by a contraction parameter $\frac{\mu}{\lambda}$, which ensures convergence toward the ground-truth parameter $\theta^*$. The incorporation of an intermediate reasoning step smooths the local optimization landscape, rendering the loss approximately convex and thereby facilitating convergence to the global optimum.

## C.4 Sample Version

We define the below sample version: assume we have the data

$$\mathcal{D}_{q,a} = \{q_i, a_i\}_{i=1}^N.$$

The loss will be:

$$L_N(\theta \mid \theta_{t-1}) = \mathbb{E}_{(q,a) \sim \widetilde{p}_{\theta^*}(\cdot)} \big[ \mathbb{E}_{z \sim \rho_\theta(\cdot \mid q)} \big[ \mathbb{E}_{(w,a') \sim \pi_\theta(\cdot \mid z,q)} [r(a', q, a)] \big] \big]$$
$$- \beta \mathbb{D}_{\text{KL}} \big( u_\theta(a', z, w \mid q) \,\|\, u_{\theta_{t-1}}(a', z, w \mid q) \big), \quad \text{(C.6)}$$

where $\widetilde{p}$ represents the empirical distribution defined as

$$\widetilde{p}(q, a) = \frac{1}{N} \sum_{i=1}^{N} 1\{(q, a) = (q_i, a_i)\}.$$

We also have the similar convergence property. Similar to the population version, we define the sample-based operator $M_N : \Theta \to \Theta$,

$$M_N(\theta) = \arg\max_{\theta' \in \Theta} L_N(\theta' \mid \theta).$$

For a given sample size $N$ and tolerance parameter $\epsilon \in (0, 1)$, define $\zeta_M^{\mathrm{unif}}(N, \epsilon)$ as the smallest scalar such that

$$\sup_{\theta \in B_2(r;\theta^*)} \|M_N(\theta) - M(\theta)\|_2 \leq \zeta_M^{\mathrm{unif}}(N, \epsilon) \tag{C.7}$$

with probability at least $1 - \epsilon$.

**Proposition C.2** (Sample Version)**.** Suppose that for all $\theta \in B(r, \theta^*)$, the mapping $M$ satisfies

$$\|M(\theta_{t-1}) - \theta^*\|_2 \leq \frac{\mu}{\lambda} \|\theta_{t-1} - \theta^*\|_2$$

with probability at least $1 - \epsilon$. Then we have

$$\|M_N(\theta_{t-1}) - \theta^*\|_2 \leq \frac{\mu}{\lambda} \|\theta_{t-1} - \theta^*\|_2 + \zeta_M^{\mathrm{unif}}(N, \epsilon), \quad \text{for all } \theta \in B(r, \theta^*)$$

with probability at least $1 - \epsilon$.

*Proof.* The result follows directly from the triangle inequality:

$$\|M_N(\theta_{t-1}) - \theta^*\|_2 \leq \|M(\theta_{t-1}) - \theta^*\|_2 + \|M_N(\theta_{t-1}) - M(\theta_{t-1})\|_2$$
$$\leq \frac{\mu}{\lambda} \|\theta_{t-1} - \theta^*\|_2 + \zeta_M^{\mathrm{unif}}(N, \epsilon).$$

$\square$

## C.5   On the Equivalence with DPO

In the deterministic setting - where $m$ or $m'$ is fixed and the responses with the maximum and minimum rewards are selected - depending on the data tuple $\big((a^{\mathrm{max}}, z^{\mathrm{max}}, w^{\mathrm{max}}), (a^{\mathrm{min}}, z^{\mathrm{min}}, w^{\mathrm{min}}), a, q\big)$, we note that in practice the construction of positive and negative samples can vary, some containing $z$ or $(z, w)$, and others including full triples such as $(a', z, w)$. For simplicity, we unify the representation and consider the minimal component shared across all formats, namely the tuple $(a', z, w)$. This process can thus be interpreted as observing a finite dataset:

$$\mathcal{D} = \{((a_i^+, z_i^+, w_i^+), (a_i^-, z_i^-, w_i^-), a_i, q_i)\}_{i=1}^{N},$$

Then the DPO loss will be:

$$L_{\mathrm{mDPO}}(\theta \mid \theta_{t-1}) = -\mathbb{E}_{((a^+, z^+, w^+),(a^-, z^-, w^-), a, q) \sim \mathcal{D}}$$
$$\log \sigma \left( \beta_{\mathrm{mDPO}} \left[ \log \frac{u_\theta(a^+, z^+, w^+ \mid q)}{u_{\theta_{t-1}}(a^+, z^+, w^+ \mid q)} - \log \frac{u_\theta(a^-, z^-, w^- \mid q)}{u_{\theta_{t-1}}(a^-, z^-, w^- \mid q)} \right] \right) \tag{C.8}$$

To demonstrate the closeness between the loss (C.6) and the loss (C.8), we first show that, with high probability, optimizing the loss (C.8) over the dataset $\mathcal{D}$ is equivalent to maximizing the original reward up to a scaling factor.

Specifically, we can derive a closed-form solution for Equation (C.6) at step $t$:

$$u_{\theta_t^*}(a', z, w \mid q) \propto u_{\theta_{t-1}}(a', z, w \mid q) \exp\left(\frac{1}{\beta} r(a', q, a)\right), \tag{C.9}$$

where $\theta_t^*$ denotes the ground truth parameter at step $t$. Accordingly, the reward function $r(a', q, a)$ can be written as $r_{\gamma_t^*}(a', q, a)$ to emphasize its dependence on the true reward parameter $\gamma_t^*$. Specifically, consider the dataset $\mathcal{D}$, which follows the following deterministic model:

$$\mathbb{P}((a^+, z^+, w^+) \succ (a^-, z^-, w^-) \mid q) = 1 \quad \text{if} \quad r_{\gamma_t^*}(a_i^+, q, a) > r_{\gamma_t^*}(a_i^-, q, a), \qquad \text{(C.10)}$$

indicating that we always select $a_i^+$ as the positive sample. To approximate this deterministic behavior, we introduce the $\alpha$-BT model:

$$\mathbb{P}((a^+, z^+, w^+) \succ (a^-, z^-, w^-) \mid q) = \frac{e^{\alpha r_{\gamma_t^*}(a^+, q, a)}}{e^{\alpha r_{\gamma_t^*}(a^+, q, a)} + e^{\alpha r_{\gamma_t^*}(a^-, q, a)}}. \qquad \text{(C.11)}$$

As $\alpha \to \infty$, the $\alpha$-BT model becomes close to the deterministic model (C.10). Then given the above dataset $\mathcal{D}$, we define the following data set

$$\mathcal{D}_\alpha = \{((a_{\alpha,i}^+, z_{\alpha,i}^+, w_{\alpha,i}^+), (a_{\alpha,i}^-, z_{\alpha,i}^-, w_{\alpha,i}^-), a_i, q_i)\}_{i=1}^n,$$

where $((a_{\alpha,i}^+, z_{\alpha,i}^+, w_{\alpha,i}^+), (a_{\alpha,i}^-, z_{\alpha,i}^-, w_{\alpha,i}^-))$ is generated according to the $\alpha$-BT model (C.11). To ensure the closeness between the dataset $\mathcal{D}$ and $\mathcal{D}_\alpha$, we have the following lemma:

*assumption* C.5 (Reward Seperation Condition). Assume that given $(q, a)$, for any $((a^+, z^+, w^+), (a^-, z^-, w^-))$, there exists $\delta$ such that $|r_{\gamma_t^*}(a^+, q, a) - r_{\gamma_t^*}(a^-, q, a)| \geq \delta$.

**Lemma C.1.** Suppose the Assumption C.5 holds, given $\epsilon$, there exists $\alpha_0 \gtrsim \frac{\log \frac{N}{2\epsilon}}{\delta}$,

$$\mathbb{P}(\mathcal{D} = \mathcal{D}_{\alpha_0}) \geq 1 - \frac{\epsilon}{2}. \qquad \text{(C.12)}$$

*Proof.* We start by bounding the probability of disagreement between two actions:

$$\mathbb{P}((a_{\alpha,i}^+, z_{\alpha,i}^+, w_{\alpha,i}^+) \neq (a_i^+, z_i^+, w_i^+)) = \frac{e^{\alpha r_{\gamma_t^*}(a^-, q, a)}}{e^{\alpha r_{\gamma_t^*}(a^+, q, a)} + e^{\alpha r_{\gamma_t^*}(a^-, q, a)}} \leq \frac{1}{1 + e^{\alpha_0 \delta}}.$$

The total probability that the datasets $\mathcal{D}$ and $\mathcal{D}_\alpha$ differ is bounded by

$$\mathbb{P}(\mathcal{D} \neq \mathcal{D}_\alpha) = \sum_{i=1}^N \mathbb{P}(a_i^+ \neq a_i^-) \leq \frac{N}{1 + e^{\alpha_0 \delta}}.$$

Given $\alpha_0 \gtrsim \frac{\log \frac{N}{2\epsilon}}{\delta}$, we conclude that

$$\mathbb{P}(\mathcal{D} = \mathcal{D}_\alpha) = 1 - \mathbb{P}(\mathcal{D} \neq \mathcal{D}_\alpha) \geq 1 - \frac{\epsilon}{2}.$$

$\square$

we can take our data generated according to the $\alpha_0$-BT model. In this case the new reward will be

$$\widetilde{r}_{\alpha_0, \gamma_t^*}(a', q, a) = \alpha_0 r_{\gamma_t^*}(a', q, a). \qquad \text{(C.13)}$$

Under this model, the minimizer of the loss (C.8) can be obtained via a two-step optimization procedure [55]:

Step 1: minimize the negative log-likelihood to obtain the reward:

$$L_{N,\text{NLL}}(\gamma \mid \alpha_0) = -\mathbb{E}_{((a^+, z^+, w^+), (a^-, z^-, w^-), a, q) \sim \mathcal{D}}$$
$$\log \sigma \big( \widetilde{r}_{\alpha_0, \gamma}(a^+, q, a) - \widetilde{r}_{\alpha_0, \gamma}(a^-, q, a) \big), \quad \text{(C.14)}$$

Denote the minimizer as $\widetilde{r}_{\alpha_0, \widehat{\gamma}_{N,t}}(a', q, a)$.
Step 2: maximize the reward $\widetilde{r}_{\alpha_0, \widehat{\gamma}_{N,t}}(a', q, a)$:

$$L_{\text{REW}}(\theta \mid \theta_{t-1}) = \mathbb{E}_{(a,q) \sim \widetilde{p}_{\theta^*}(\cdot)} \Big[ \mathbb{E}_{z, w \sim \widetilde{p}_{\theta_{t-1}}(\cdot \mid a, q)} \big[ \mathbb{E}_{a' \sim f_\theta(\cdot \mid z, w, q, a)} [\widetilde{r}_{\alpha_0, \widehat{\gamma}_{N,t}}(a', q, a)] \big] \Big]$$
$$- \beta_{\text{mDPO}} \mathbb{D}_{\text{KL}} \big( u_\theta(a' \mid z, w, q, a) \,\|\, u_{\theta_{t-1}}(a' \mid z, w, q, a) \big). \quad \text{(C.15)}$$

The solution will be

$$u_{\widehat{\theta}_t}(a' \mid z, w, q, a) \propto u_{\theta_{t-1}}(a' \mid z, w, q, a) \exp\left(\frac{1}{\beta_{\mathrm{mDPO}}} \widetilde{r}_{\alpha_0, \widehat{\gamma}_{N,t}}(a', q, a)\right), \tag{C.16}$$

This expression is identical to Equation (C.9), except that it uses a different parameterization of the reward. Specifically, the reward function $\widetilde{r}_{\alpha_0, \gamma_t^*}(a', q, a)$ is parameterized by the ground truth $\gamma_t^*$ and a hyperparameter $\alpha_0$. To ensure uniform consistency of the maximum likelihood estimator, we invoke the following lemma, which is modified from Theorem 5.7 in [69]. This result guarantees that the minimizer in Step 1 converges to the true reward function $\widetilde{r}_{\alpha_0, \gamma_t^*}(a', q, a)$. We need the following assumption:

*assumption* C.6. Suppose that there exists a constant $c_\alpha > 0$, for every $\epsilon > 0$, such that:

$$\sup_{\alpha_0 \in [c_\alpha, \infty)} \sup_{\gamma \in \Gamma} |L_{N,\mathrm{NLL}}(\gamma \mid \alpha_0) - L_{\mathrm{NLL}}(\gamma \mid \alpha_0)| \xrightarrow{P} 0, \tag{C.17}$$

where $\Gamma$ represents the parameter space and

$$\sup_{\alpha_0 \in [c_\alpha, \infty)} \sup_{\gamma : \|\gamma - \gamma_t^*\|_2 \geq \epsilon} -(L_{\mathrm{NLL}}(\gamma \mid \alpha_0) - L_{\mathrm{NLL}}(\gamma_t^* \mid \alpha_0)) < 0. \tag{C.18}$$

**Lemma C.2** (Uniform MLE Consistency). Let $L_{N,\mathrm{NLL}}(\gamma \mid \alpha_0)$ be the negative log-likelihood function, and let $L_{\mathrm{NLL}}(\gamma \mid \alpha_0)$ denote its expected version. Let Assumption C.6 holds, then for the sequence of estimators $\widehat{\gamma}_{N,t}$ obtained form minimizing the loss (C.14), we have: given $\epsilon > 0$, there exists $N_1$, when $N \geq N_1$, for any $\alpha_0 \in [c_\alpha, \infty)$,

$$\mathbb{P}\big(\|\widehat{\gamma}_{N,t} - \gamma_t^*\|_2 \leq \epsilon\big) \geq 1 - \frac{\epsilon}{2}. \tag{C.19}$$

*Proof.* For given $\epsilon$, according to the Equation (C.18), there exists $c_{\epsilon, NLL}$, such that:

$$\sup_{\alpha_0 \in [c_\alpha, \infty)} \sup_{\gamma : \|\gamma - \gamma_t^*\|_2 \geq \epsilon} -(L_{\mathrm{NLL}}(\gamma \mid \alpha_0) - L_{\mathrm{NLL}}(\gamma_t^* \mid \alpha_0)) < -c_{\epsilon, NLL}.$$

For $c_{\epsilon, NLL}$, according to Equation (C.17), there exists $N_1$, when $N \geq N_1$, for any $\alpha_0 \in [c_\alpha, \infty)$,

$$\mathbb{P}\left(|L_{N,\mathrm{NLL}}(\widehat{\gamma}_{N,t} \mid \alpha_0) - L_{\mathrm{NLL}}(\widehat{\gamma}_{N,t} \mid \alpha_0)| \leq \frac{c_{\epsilon, NLL}}{3}\right) \geq 1 - \frac{\epsilon}{4},$$

$$\mathbb{P}\left(|L_{N,\mathrm{NLL}}(\gamma_t^* \mid \alpha_0) - L_{\mathrm{NLL}}(\gamma_t^* \mid \alpha_0)| \leq \frac{c_{\epsilon, NLL}}{3}\right) \geq 1 - \frac{\epsilon}{4}.$$

Since $\widehat{\gamma}_{N,t}$ is the minimizer of loss (C.14), for any $\alpha_0 \in [c_\alpha, \infty)$, we have:

$$L_{N,\mathrm{NLL}}(\widehat{\gamma}_{N,t} \mid \alpha_0) \leq L_{N,\mathrm{NLL}}(\gamma_t^* \mid \alpha_0)$$

Consequently,

$$\mathbb{P}\left(-(L_{\mathrm{NLL}}(\gamma \mid \alpha_0) - L_{\mathrm{NLL}}(\gamma_0 \mid \alpha_0)) \geq -\frac{2c_{\epsilon, NLL}}{3}\right) \geq 1 - \frac{\epsilon}{2}.$$

Thus for any $\alpha_0 \in [c_\alpha, \infty)$,

$$\mathbb{P}\big(\|\widehat{\gamma}_{N,t} - \gamma_t^*\|_2 \leq \epsilon\big) \geq \mathbb{P}\left(-(L_{\mathrm{NLL}}(\gamma \mid \alpha_0) - L_{\mathrm{NLL}}(\gamma_0 \mid \alpha_0)) \geq -\frac{2c_{\epsilon, NLL}}{3}\right) \geq 1 - \frac{\epsilon}{2}.$$

$\square$

Having established the necessary groundwork, we are now ready to present Theorem C.2, which establishes the equivalence between the minimizes of the two loss functions:

**Theorem C.2.** Assume Assumptions C.5 and C.6 hold, given $\epsilon$, there exists $N_1$ and $\beta_{\mathrm{mDPO}} \gtrsim \frac{\log \frac{N_1}{2\epsilon}}{\delta} \beta$, the minimizer of loss (C.8) $\widehat{\theta}_{t,\mathrm{mDPO}}$ will satisfy:

$$\mathbb{P}\left(\left\|\widehat{\theta}_{t,\mathrm{mDPO}} - \theta_t^*\right\|_2 \geq \epsilon\right) < \epsilon, \tag{C.20}$$

where $\theta_t^*$ is defined in Equation (C.9).

*Proof.* First, according to the Lemma C.2, there exists $N_1$, if we define the event $\Omega_1 = \{\|\widehat{\gamma}_{N_1,t} - \gamma_t^*\|_2 \le \epsilon\}$, , we have:

$$\mathbb{P}(\Omega_1) \ge 1 - \frac{\epsilon}{2}.$$

Secondly, just choose the sample size of $\mathcal{D}$ as $NK$, define the event $\Omega_2 = \{\mathcal{D} = \mathcal{D}_{\alpha_0}\}$, by Lemma C.1, when we take $\alpha_0 \gtrsim \left(\frac{\log \frac{N_1}{2\epsilon}}{\delta} \vee c_\alpha\right)$, we have:

$$\mathbb{P}(\Omega_2) \ge 1 - \frac{\epsilon}{2}.$$

Since $c_\alpha$ is a constant, we may, without loss of generality, take $\alpha_0 \gtrsim \frac{\log \frac{N_1}{2\epsilon}}{\delta}$. Henceforth, we restrict our analysis to the event $\Omega_1 \cap \Omega_2$, which occurs with probability at least $1 - \epsilon$. Conditioned on this event, the data can be viewed as being generated from the $\alpha_0$-BT model (C.11). Consequently, the minimizer of the loss (C.8) coincides with that of Equation (C.16):

$$u_{\widehat{\theta}_{t,\text{mDPO}}}(a', z, w \mid q) \propto u_{\theta_{t-1}}(a', z, w \mid q) \exp\left(\frac{1}{\beta_{\text{mDPO}}} \widetilde{r}_{\alpha_0, \widehat{\gamma}_{N_1,t}}(a', q, a)\right)$$

$$\propto u_{\theta_{t-1}}(a', z, w \mid q) \exp\left(\frac{\alpha_0}{\beta_{\text{mDPO}}} r_{\widehat{\gamma}_{N_1,t}}(a', q, a)\right).$$

Compared to the solution in (C.9), when $\beta_{\text{mDPO}} = \alpha_0\beta \gtrsim \frac{\log \frac{N_1}{2\epsilon}}{\delta}\beta$, controlling the distance between $\widehat{\theta}t$, mDPO and $\theta_t^*$ reduces to controlling the distance between $\widehat{\gamma}N_1, t$ and $\gamma_t^*$, as established by Lemma C.2. Consequently, we obtain:

$$\mathbb{P}\left(\left\|\widehat{\theta}_{t,\text{mDPO}} - \theta_t^*\right\|_2 \ge \epsilon\right) < \epsilon.$$

This concludes the proof. $\qquad\square$

## C.6 Convergence Property of DPO

Finally, combining Proposition C.2, we conclude that the sequence $\widehat{\theta}_{t,\text{mDPO}}$ converges as $t$ increases. We formally state the following theorem:

**Theorem C.3.** For a given iteraton number $T$, for some radius $r > 0$ and pair $(\mu, \lambda)$ such that $0 \le \mu < \lambda$, suppose that the Assumption C.1-C.6 hold and assume $(\epsilon + \zeta_M^{\text{unif}}(N, \epsilon)) < (1 - \frac{\mu}{\lambda})r$, then with probability at least $1 - (T+1)\epsilon$, we have:

$$\left\|\widehat{\theta}_{T,\text{mDPO}} - \theta^*\right\|_2 \le \left(\frac{\mu}{\lambda}\right)^T \|\theta_{\text{ref}} - \theta^*\|_2 + \frac{1}{1 - \frac{\mu}{\lambda}} \zeta_M^{\text{unif}}(n, \epsilon)$$

*Proof.* Notice that $\theta_t^* = M_N(\widehat{\theta}_{t-1,\text{mDPO}})$, apply Proposition C.2, we get:

$$\|\theta_t^* - \theta^*\|_2 \le \frac{\mu}{\lambda}\left\|\widehat{\theta}_{t-1,\text{mDPO}} - \theta^*\right\|_2 + \zeta_M^{\text{unif}}(N, \epsilon)$$

with probability at least $1 - \epsilon$. Combining Theorem C.2,

$$\left\|\widehat{\theta}_{t,\text{mDPO}} - \theta^*\right\|_2 \le \frac{\mu}{\lambda}\left\|\widehat{\theta}_{t-1,\text{mDPO}} - \theta^*\right\|_2 + \epsilon + \zeta_M^{\text{unif}}(N, \epsilon),$$

with probability at least $1 - 2\epsilon$. Notice that $(\epsilon + \zeta_M^{\text{unif}}(N, \epsilon)) \le (1 - \frac{\mu}{\lambda})r$, then $\widehat{\theta}_{t,\text{mDPO}} \in B(r, \theta^*)$. Based on this, we can perform iteration:

$$\left\|\widehat{\theta}_{T,\text{mDPO}} - \theta^*\right\|_2 \le \frac{\mu}{\lambda}\left\|\widehat{\theta}_{T-1,\text{mDPO}} - \theta^*\right\|_2 + \epsilon + \zeta_M^{\text{unif}}(N, \epsilon)$$

$$\le \frac{\mu}{\lambda}\left(\frac{\mu}{\lambda}\left\|\widehat{\theta}_{T-2,\text{mDPO}} - \theta^*\right\|_2 + \epsilon + \zeta_M^{\text{unif}}(N, \epsilon)\right)$$

$$\le \left(\frac{\mu}{\lambda}\right)^T \|\theta_{\text{ref}} - \theta^*\|_2 + \sum_{s=0}^{T-1} \left(\frac{\mu}{\lambda}\right)^s (\epsilon + \zeta_M^{\text{unif}}(N, \epsilon))$$

$$\le \left(\frac{\mu}{\lambda}\right)^T \|\theta_{\text{ref}} - \theta^*\|_2 + \frac{1}{1 - \frac{\mu}{\lambda}}(\epsilon + \zeta_M^{\text{unif}}(N, \epsilon))$$

with probability at least $1 - (T+1)\epsilon$. $\qquad\square$

# D Information for Test Datasets

The information of the test datasets used in AceSearcher is listed in the following table. Note that We conduct evaluations on all questions from StrategyQA and Bamboogle, and the first 500 questions from the development sets of the other datasets following existing studies [68, 57, 36]. For dataset in DocMathEval, we use the `testmini` version as the evaluation set to compare the performance of AceSearcher and baselines.

Table 4: Descriptions of datasets used in AceSearcher. For SimpLong and CompLong, we use `text-embedding-3` to retrieve top-10 relevant context before generate the answer.

| Dataset | Description |
|---------|-------------|
| 2WikiMHQA [21] | 2WikiMultiHopQA is a multi-hop question answering dataset built from Wikipedia, where each question requires reasoning over two distinct articles. It emphasizes information synthesis across multiple documents for accurate answer retrieval. |
| HotpotQA [83] | HotpotQA is a crowd-sourced multi-hop QA dataset where each question demands reasoning over multiple Wikipedia passages. It also includes supporting fact annotations to promote explainability in QA systems. |
| Bamboogle [53] | Bamboogle is a multi-hop QA dataset constructed using Bing search engine snippets. It presents naturally occurring, challenging questions requiring reasoning over diverse web snippets rather than structured sources like Wikipedia. |
| MusiQue [67] | MusiQue is a multi-hop QA dataset featuring real-world questions from community forums like Quora and Yahoo Answers. It targets complex questions requiring synthesis across multiple evidence passages, each carefully annotated. |
| HOVER [28] | HOVER is a multi-hop QA dataset with annotated supporting facts, built on entity-linked Wikipedia documents. It stresses explainable reasoning by providing intermediate evidence chains. |
| ExFEVER [47] | ExFEVER extends the FEVER dataset by introducing multi-hop claims requiring evidence from multiple documents. It is designed to support research on fact verification and evidence-based reasoning. |
| $DM_{SS}$ (DocMath SimpShort) | A dataset reannotated from TAT-QA [94] and FinQA [7], consisting of short financial documents with a single table for simple numerical reasoning. |
| $DM_{CS}$ (DocMath CompShort) | A dataset reannotated from TAT-HQA [35], consisting of short single-table documents for complex numerical reasoning, including hypotheticals. |
| $DM_{SL}$ (DocMath SimpLong) | A dataset reannotated from MultiHiertt [90], consisting of long multi-table financial documents for simple reasoning in realistic contexts. |
| $DM_{CL}$ (DocMath CompLong) | A dataset of long, structured financial documents requiring multi-step compositional numerical reasoning. |

# E   Details of Training Data

We provide the data composition for SFT and RFT, including their corresponding tasks, links to access the data, and the number we use in each stage in Table 5. To avoid data contamination, we follow the instructions in the MusiQue repository and *remove* the training data with overlapping IDs from NQ and Squad to avoid data leakage.

Table 5: The data composition for SFT and RFT stages.

| Dataset | Task | Link | Count |
|---|---|---|---|
| *Data composition for SFT* | | | |
| NarrativeQA [32] | Context-rich QA | https://huggingface.co/datasets/deepmind/narrativeqa | 20000 |
| SQuAD 1.1 [56] | Context-rich QA | https://rajpurkar.github.io/SQuAD-explorer/ | 10000 |
| SQuAD 2.0 [56] | Context-rich QA | https://rajpurkar.github.io/SQuAD-explorer/ | 10000 |
| TAT-QA [94] | Context-rich QA | https://github.com/NExTplusplus/TAT-QA/tree/master/dataset_raw | 12000 |
| FEVER [66] | Context-rich QA | https://fever.ai/dataset/fever.html | 10000 |
| DROP [16] | Context-rich QA | https://huggingface.co/datasets/ucinlp/drop | 20000 |
| Quoref [12] | Context-rich QA | https://huggingface.co/datasets/allenai/quoref | 20000 |
| ROPES [38] | Context-rich QA | https://huggingface.co/datasets/allenai/ropes | 10000 |
| NQ [33] | Context-rich QA | https://dl.fbaipublicfiles.com/dpr/data/retriever/biencoder-nq-train.json.gz | 20000 |
| GSM8K [11] | Question Decomposition | https://huggingface.co/datasets/openai/gsm8k/viewer/socratic | 7000 |
| ConvFinQA [9] | Question Decomposition | https://github.com/czyssrs/ConvFinQA | 1000 |
| StrategyQA [19] | Question Decomposition | https://huggingface.co/datasets/ChilleD/StrategyQA | 1600 |
| IfQA [85] | CoT | https://github.com/wyu97/IfQA/tree/main/dataset | 2000 |
| TabMWP [46] | CoT | https://promptpg.github.io/index.html#dataset | 10000 |
| GSM8K [11] | CoT | https://huggingface.co/datasets/openai/gsm8k/viewer/socratic | 7000 |
| MathInstruct-COT [87] | CoT | https://huggingface.co/datasets/TIGER-Lab/MathInstruct | 10000 |
| MathInstruct-POT [87] | CoT | https://huggingface.co/datasets/TIGER-Lab/MathInstruct | 10000 |
| **TOTAL** | — | — | **180600** |
| *Data composition for RFT* | | | |
| HotpotQA [83] | RAG | https://github.com/hotpotqa/hotpot | 10000 |
| 2WikiMQA [21] | RAG | https://huggingface.co/datasets/xanhho/2WikiMultihopQA | 10000 |
| HOVER [28] | RAG | https://github.com/hover-nlp/hover | 10000 |
| GSM8K [11] | Context-rich Reasoning | https://huggingface.co/datasets/openai/gsm8k/viewer/socratic | 7000 |
| TabMWP [46] | Context-rich Reasoning | https://promptpg.github.io/index.html#dataset | 10000 |
| ConvFinQA [9] | Context-rich Reasoning | https://github.com/czyssrs/ConvFinQA | 2000 |
| **Total** | — | — | **49000** |

# F   Prompt Templates

## F.1   Prompts for Direct RAG

You have the following context passages: {context}

Given the question: "{question}" as well as the context above, please answer the above question with one or a list of entities with the given context as the reference. Your answer needs to be a span with one or a list of entities.

Figure 6: Prompt for direct RAG on complex question answering tasks.

Answer the following questions with SUPPORTED or NOT_SUPPORTED with the given context as the reference.

**Question:** {question}
**Context:** {context}

Your answer should only be SUPPORTED or NOT_SUPPORTED.

Figure 7: Prompt for direct RAG on fact verification tasks.

You have the following passages and table:

**Passages:**
{passage}

**Tables:**
{table}

For the question "{question}", write a Python program to solve the question. Store the final result in the variable ans.

Figure 8: Prompt for direct RAG on document-level reasoning tasks with PoT.

You have the following passages and table:

**Passages:**
{passage}

For the question "{question}", reason step by step to calculate the final answer. Please use \boxed{} to wrap your final answer.

Figure 9: Prompt for direct RAG on document-level reasoning tasks with CoT.

## F.2 Prompts for Decomposition

Please break down the question "{question}" into multiple specific sub-questions that address individual components of the original question.

Mark each sub-question with ### at the beginning. If you need to refer to answers from earlier sub-questions, use #1, #2, etc., to indicate the corresponding answers.

**Decomposed question:**

Figure 10: Prompt for question decomposition on complex question answering tasks.

Please break down the claim "{claim}" into multiple smaller sub-claims that each focus on a specific component of the original statement, making it easier for a model to verify. Begin each sub-claim with ###. If needed, refer to answers from earlier sub-claims using #1, #2, etc.

**Decomposed claim:**

Figure 11: Prompt for question decomposition on fact verification tasks.

You have the following passages and table:

**Passages:**
{passages}

**Tables:**
{tables}

Please break down the question "{question}" into multiple specific sub-questions that address individual components of the original question, with the table and passages as the reference. Use ### to mark the start of each sub-question.

**Decomposed question:**

Figure 12: Prompt for question decomposition on document-level reasoning tasks.

## F.3 Prompts for subquestion answering

You have the following context passages:
{passages}

Please answer the question "{subquestion}" with a short span using the context as reference. If no answer is found in the context, use your own knowledge. Your answer needs to be as short as possible.

Figure 13: Prompt for subquestion answering on complex question answering tasks.

You have the following context passages:
{passages}

Please verify whether the claim "{subquestion}" is correct using the context as reference. If no answer is found in the context, use your own knowledge. Please only output Yes or No and do not give any explanation.

Figure 14: Prompt for subquestion answering on fact verification tasks.

You have the following passages and tables:

**Passage:**
{passages}

**Table:**
{tables}

For the question "{subquestion}", write a Python program to solve the question. Store the final result in the variable ans.

Figure 15: Prompt for subquestion answering on document-level reasoning tasks with PoT.

You have the following passages and tables:

**Passage:**
{passages}

**Table:**
{tables}

For the question "{subquestion}", reason step by step to calculate the final answer. Please use \boxed{} to wrap your final answer.

Figure 16: Prompt for subquestion answering on document-level reasoning tasks with CoT.

## F.4   Prompts for final answer generation

You have the following passages:
{passages}

You are also given some subquestions and their answers:
**# subquestion #1:** {subquestion_1} **Answer:** {answer_1}
**# subquestion #2:** {subquestion_2} **Answer:** {answer_2}
· · ·

Please answer the question "{the_original_question}" with a short span using the documents and subquestions as reference.

Make sure your response is grounded in documents and provides clear reasoning followed by a concise conclusion. If no relevant information is found, use your own knowledge.

Wrap your answer with <answer> and </answer> tags.

Figure 17: Prompt for final answer generation on complex question answering tasks.

You are given some subquestions and their answers:
**# subquestion #1:** {subquestion_1} **Answer:** {answer_1}
**# subquestion #2:** {subquestion_2} **Answer:** {answer_2}
. . .

Please answer the question "{the_original_question}" with only Yes or No using the subquestions as reference. Provides clear reasoning followed by a concise conclusion. If no relevant information is found, use your own knowledge.

Wrap your answer with <answer> and </answer> tags.

Figure 18: Prompt for final answer generation on fact verification tasks.

You have the following passages and table:

**Passages:**
{passage}

For the question "{question}", here is a referenced breakdown:
{decomposition}.

Write a Python program to solve the question. Store the final result in the variable ans.

Figure 19: Prompt for final answer generation on document-level reasoning tasks with PoT.

You have the following passages and table:

**Passages:**
{passage}

For the question "{question}", here is a referenced breakdown:
{decomposition}.

Reason step by step to calculate the final answer. Please use \boxed{} to wrap your final answer.

Figure 20: Prompt for final answer generation on document-level reasoning tasks with CoT.

### F.5 Prompts for InstructRAG

Read the following documents relevant to the given question: {question}

**Documents:**
{documents}
...

Please identify documents that are useful to answer the given question: "{question}". If none of the documents is aligned with the answer, in that case, you have to explain the answer only based on your own knowledge, without referring to the provided information.

Note that the question may be compositional and require intermediate analysis to deduce the final answer. Make sure your response is grounded and provides clear reasoning details followed by a concise conclusion. Your answer should be in a short span with a few keywords. Use <answer> and </answer> tag to mark your final answer.

Figure 21: Prompt for InstructRAG on complex question answering tasks.

Read the following documents relevant to the given question: {question}

**Documents:**
{documents}
...

Please identify documents that are useful to answer the given question: "{question}". If none of the documents is aligned with the answer, in that case, you have to explain the answer only based on your own knowledge, without referring to the provided information.

Note that the question may be compositional and require intermediate analysis to deduce the final answer. Make sure your response is grounded and provides clear reasoning details followed by a concise conclusion. Your answer should be yes or no only. Use <answer> and </answer> tag to mark your final answer.

Figure 22: Prompt for InstructRAG on fact verification tasks.

## G Additional Implementation Details

### G.1 Implementation Details for SFT

For SFT, we set the batch size to 64 for every example, and set the learning rate as Table 7. With maximum number of tokens to 2560.

Table 6: Results for different model sizes for SFT.

| Model Size | Learning Rate | Warmup Steps |
|---|---|---|
| AceSearcher 1.5B | $5e - 6$ | 5% |
| AceSearcher 8B | $1e - 6$ | 5% |
| AceSearcher 14B | $1e - 6$ | 5% |
| AceSearcher 32B (LoRA) | $1e - 5$ | 5% |

### G.2 Implementation Details for RFT

We set the hyperparameters to $m = 3$, $m' = 4$, and $t = 1.0$ when generating multiple rollouts. Examples with identical maximum and minimum rewards are discarded. For RFT, we use $\beta = 0.1$

Table 7: Results for different model sizes for RFT.

| Model Size | Learning Rate | Warmup Steps |
|---|---|---|
| AceSearcher 1.5B | $1e-6$ | 5% |
| AceSearcher 8B | $5e-7$ | 5% |
| AceSearcher 14B | $5e-7$ | 5% |
| AceSearcher 32B (LoRA) | $1e-6$ | 5% |

and run for the DPO for 2 iterations by default. All models are optimized using AdamW with $\beta_1 = 0.9$ and $\beta_2 = 0.98$, and experiments are conducted on 8 NVIDIA A100 GPUs.

### G.3 Implementation Details for Baselines

We implement and evaluate a variety of baselines using standardized decoding and prompting configurations to ensure fair comparison. For **Qwen-3**, we follow the official guidance[5] to adopt distinct sampling strategies depending on the task setting. In *thinking mode* (`enable_thinking=True`), we use temperature = 0.6, top-p = 0.95, top-k = 20, and min-p = 0 to encourage diverse yet coherent generation. Greedy decoding is explicitly avoided to prevent performance degradation and repetitive outputs. In *non-thinking mode* (`enable_thinking=False`), we slightly increase the temperature to 0.7 and reduce top-p to 0.8 while keeping top-k and min-p unchanged. In practice, we find that using the thinking mode leads to slightly better performance despite being slower. For **R1-distill** models, we set the maximum generation length to 32,768 tokens and use temperature = 0.6, top-p = 0.95. In **Plan-RAG**, we incorporate 3-shot demonstrations in the prompt to guide the model toward producing outputs in the correct format. For **InstructRAG**, we use the same SFT training set as AceSearcher and generate CoT-style demonstrations tailored to context-rich QA datasets. For **Llama-4**, **GPT-4.1**, and **GPT-4o**, we use greedy decoding (temperature = 0) for consistency with their default inference behavior. For **IRCOT** and **RAG-Star**, we reproduce results by following the original repositories and hyperparameter settings. For these methods, we tune the number of retrieved passages from $\{5, 10, 20\}$ and report the best performance. We refer to other baselines' reported numbers in the corresponding paper.

## H Additional Experimental Results

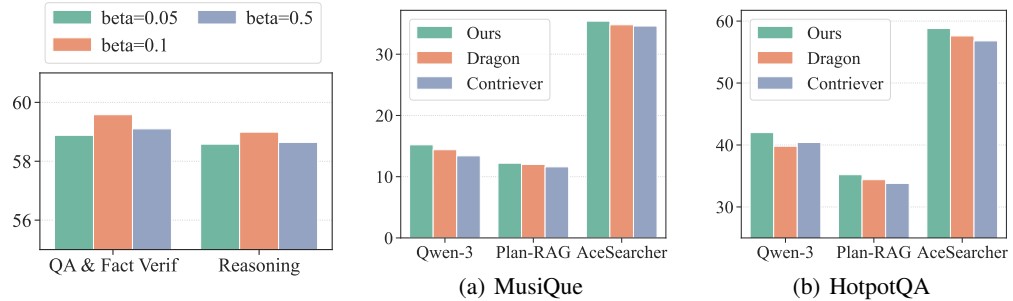

Figure 23: Parameter Study on $\beta$

Figure 24: Effect of different retrievers.

**Effect of $\beta$.** We study the effect of $\beta$ in preference optimization with Llama-3.1-8B as the backbone, and find that AceSearcher is generally robust to this parameter, with $\beta = 0.1$ leads to slightly better performance.

**Effect of Different Retrievers.** We evaluate AceSearcher and representative baselines (at the 8B scale) using two different retrievers: Dragon[6] and Contriever[7]. Overall, the E5 retriever achieves

---

[5] https://huggingface.co/Qwen/Qwen3-32B#best-practices
[6] https://huggingface.co/facebook/dragon-plus-context-encoder
[7] https://huggingface.co/facebook/contriever-msmarco

Table 8: Performance comparison across models and prompting methods.

| Model | Prompt Method | DM$_{SimpShort}$ | DM$_{CompShort}$ | DM$_{SimpLong}$ | DM$_{CompLong}$ | Avg. |
|---|---|---|---|---|---|---|
| AceSearcher-32B | PoT | 89.5 | 84.0 | 53.0 | 43.0 | 66.1 |
| AceSearcher-14B | PoT | 84.0 | 82.0 | 49.0 | 39.3 | 62.4 |
| AceSearcher-8B | PoT | 83.0 | 80.5 | 48.0 | 32.3 | 59.0 |
| AceSearcher-1.5B | PoT | 66.5 | 77.5 | 39.0 | 18.0 | 47.6 |
| AceSearcher-32B | CoT | 73.5 | 70.0 | 50.0 | 33.0 | 54.5 |
| AceSearcher-14B | CoT | 78.5 | 75.5 | 44.0 | 34.7 | 57.0 |
| AceSearcher-8B | CoT | 44.0 | 31.5 | 30.0 | 15.7 | 28.5 |
| AceSearcher-1.5B | CoT | 37.5 | 32.0 | 18.0 | 9.7 | 23.2 |

the best performance, supporting our hypothesis that stronger retrieval models yield more relevant passages and thus enhance answer quality. Notably, AceSearcher consistently outperforms baselines across different retrievers, demonstrating its robustness to retrieval choices.

**Comparison of CoT and PoT for Document-level Reasoning.** Table 8 presents a comparison between Program of Thought (POT) and Chain of Thought (COT) prompting methods across four evaluation settings. POT consistently outperforms COT across all tasks, with notable improvements on both simple and complex reasoning benchmarks. For example, across models, POT yields higher average scores than COT on DM$_{CompLong}$ (e.g., 43.0 vs. 33.0 for AceSearcher-32b) and DM$_{SimpShort}$ (e.g., 89.5 vs. 73.5 for AceSearcher-32b), demonstrating its advantage in guiding structured reasoning. These results highlight the effectiveness of POT in enhancing model performance on decision-making tasks requiring multi-step reasoning, regardless of model scale.

# I   Case Studies

**Details of Human Study** Our human study has received IRB approval from our institute. The below is the form used in human evaluation:

You are provided with:

- The original complex question
- A proposed decomposition into subquestions

Please assess the overall quality of the decomposition using the criteria below.

- Relevance: Do the subquestions help solve the original question?
- Completeness: Are all important aspects covered?
- Coherence: Is the breakdown logically structured and easy to follow?
- Usefulness: Does the decomposition make the reasoning process easier or more interpretable?

Please rate the overall quality of the subquestion decomposition.

- 1: Very Poor: Subquestions are irrelevant, incomplete, or incoherent.
- 2: Poor: Some relevance, but major gaps or unclear logic.
- 3: Fair: Moderately helpful with minor issues in coverage or clarity.
- 4: Good: Clear and mostly complete decomposition.
- 5: Excellent: Subquestions are highly relevant, complete, and well-structured, with proper reference to answers from previous subquestions.

**Details of Case Study**

We present two case studies from the MusiQue, 2WikiMHQA and DocMathEval datasets in Tables 9, 10 and 11. Despite its larger size, the Qwen3-32B-reasoning model struggles with handling complex

questions. In contrast, AceSearcher effectively decomposes each question into manageable sub-questions, generates accurate answers for each sub-questions, and performs multi-step reasoning to arrive at the correct final answer.

Table 9: A case study from MusiQue dataset.

| A sample from MusiQue |
| --- |
| **Question:** In which state is Vera Barbosa's place of birth located?
**Ground Truth Answer:** Lisbon District |
| **AceSearcher-8B** |
| **Sub-question 1:** Who is Vera Barbosa?
**Answer 1:** a Portuguese track and field athlete
**Sub-question 2:** Where was Vera Barbosa born?
**Answer 2:** Vila Franca de Xira
**Sub-question 3:** In which state is #2 located?
**Answer 3:** Lisbon District

**Final Answer:**
The documents state that Vera Barbosa is a Portuguese track and field athlete and was born in Vila Franca de Xira. Additionally, Vila Franca de Xira is located in the Lisbon District. Therefore, Vera Barbosa's place of birth, Vila Franca de Xira, is in the Lisbon District.

**<answer>Lisbon District</answer>** |
| **Qwen3-32B Reasoning** |
| <think>
Okay, let's see. The question is asking about the state where Vera Barbosa's place of birth is located. First, I need to figure out which Vera Barbosa they're referring to here because there are several people named Vera Barbosa in the provided contexts.

Looking at the first entry: Vera Barbosa, born 13 January 1989 in Vila Franca de Xira, Portugal. That's a Portuguese athlete. Then there's another Vera Krasova, a Russian model, and Vera Begić, a Croatian athlete. There's also a mention of a place called Barbosa in Colombia and Brazil. But the key here is the name Vera Barbosa specifically.

The user is asking about Vera Barbosa's birthplace state. The first context says she was born in Vila Franca de Xira, which is in Portugal. However, the question is phrased as "In which state is Vera Barbosa's place of birth located?" So, states are typically administrative divisions in countries like the US, Brazil, etc. Portugal is divided into districts, not states. But maybe the user is using "state" in a general sense here.

Wait, but looking at other entries, there's a Carlos Barbosa in Brazil's Rio Grande do Sul state, and Bárbara Barbosa is a Brazilian footballer. But the specific Vera Barbosa mentioned in the first paragraph is the Portuguese athlete. Her birthplace is Vila Franca de Xira, Portugal. Since Portugal doesn't have states, but the question uses "state," perhaps the answer is Portugal. Alternatively, if the user is confused and actually refers to a different person, but based on the given data, the only Vera Barbosa with a birthplace listed is the Portuguese one. Therefore, the answer should be Portugal.
</think>

**<answer>Portugal</answer>** |

Table 10: A case study from 2WikiMHQA dataset.

| A sample from 2WikiMHQA |
| --- |

**Question:** Which film has the director who was born later, The Silver Treasure or Taxi To Paradise?
**Ground Truth Answer:** Taxi To Paradise

| AceSearcher-8B |
| --- |

**Sub-question 1:** Who directed The Silver Treasure?
**Answer 1:** Rowland V. Lee
**Sub-question 2:** Who directed Taxi To Paradise?
**Answer 2:** Adrian Brunel
**Sub-question 3:** When was the director of #1 born?
**Answer 3:** September 6, 1891
**Sub-question 4:** When was the director of #2 born?
**Answer 4:** 4 September 1892
**Sub-question 5:** Is the year of #3 later than #4?
**Answer 5:** no

**Final Answer:**
From the documents, "The Silver Treasure" was directed by Rowland V. Lee, born on September 6, 1891 (Q3), and "Taxi To Paradise" was directed by Adrian Brunel, born on September 4, 1892 (Q4). Comparing these dates, Adrian Brunel was born later than Rowland V. Lee (Q5 confirms this).
Thus, the film with the director born later is "Taxi To Paradise."

<answer>Taxi To Paradise</answer>

| Qwen3-32B Reasoning |
| --- |

<think>
Okay, let's see. The user is asking which film has the director born later between The Silver Treasure and Taxi to Paradise.

First, I need to find the directors of both films. From the context:

- The Silver Treasure (1926) is directed by Rowland V. Lee. - Taxi to Paradise (1933) is directed by Adrian Brunel.

Now, I need to check their birth dates. The context doesn't provide birth dates for these directors. So I'll have to rely on my own knowledge.

Rowland V. Lee was born on April 22, 1894. Adrian Brunel was born on October 2, 1887. Wait, Brunel was born earlier than Lee? Wait, no: if Brunel was born in 1887 and Lee in 1894, then Brunel was born earlier. Therefore, Rowland V. Lee was born later. So The Silver Treasure's director was born later than Taxi to Paradise's director. Therefore, the answer is The Silver Treasure.
</think>

<answer>The Silver Treasure</answer>

Table 11: A case study from DM$_{CS}$ subset of the DocMathEval dataset.

**A sample from DM$_{CS}$ subset of DocMathEval.**

**Table:**

| Year | Segment | Americas | Europe | Asia | Total | % of Total |
|------|---------|----------|--------|------|-------|------------|
| | | | (Amounts in thousands) | | | |
| 2019 | TS | $67,728 | $3,285 | $646 | $71,159 | 90% |
| 2019 | HPP | $5,294 | $771 | $1,837 | $7,902 | 10% |
| 2019 | Total | $72,522 | $4,056 | $2,483 | $79,061 | 100% |
| 2019 | % of Total | 92% | 5% | 3% | 100% | |
| 2018 | TS | $52,034 | $9,059 | $1,344 | $62,437 | 86% |
| 2018 | HPP | $8,424 | $1,266 | $789 | $10,479 | 14% |
| 2018 | Total | $60,458 | $10,325 | $2,133 | $72,916 | 100% |
| 2018 | % of Total | 83% | 14% | 3% | 100% | |

Table 12: The following table details the Company's sales by operating segment for fiscal years ended September 30, 2019 and 2018. The Company's sales by geographic area based on the location of where the products were shipped or services rendered are as follows: Substantially all Americas amounts are United States.

**Question:** What is the percentage change in Asia sales between 2018 and 2019 if the 2019 sales is doubled and increased by another 400 thousand? (in percent)
**Ground Truth Answer:** 151.5705

**AceSearcher-8B**

**Sub-question 1:** What was the value of Asia sales in 2019?
**Sub-question 2:** What is the value of Asia sales in 2019 after doubling it?
**Sub-question 3:** What is the value of Asia sales in 2019 after doubling it and then adding 400,000?
**Sub-question 4:** What was the value of Asia sales in 2018?
**Sub-question 5:** What is the net change in the value of Asia sales from 2018 to 2019, after adjusting the 2019 value?
**Sub-question 6:** What is the percentage change in the value of Asia sales from 2018 to 2019, after adjusting the 2019 value?

**Final Answer:**
```
# Given data
asia_sales_2019 = 2483 # in thousands (Q1)
asia_sales_2018 = 2133 # in thousands (Q4)

# Adjust 2019 sales:  double it and add 400 thousand (Q2, Q3)
adjusted_2019_sales = (asia_sales_2019 * 2) + 400

# Compute value change (Q5)
change_in_sales = adjusted_2019_sales - asia_sales_2018

# Compute percentage change (Q6)
ans = (change_in_sales / asia_sales_2018) * 100
```

151.57055789967183

**Qwen3-32B Reasoning**

```
# Original 2019 Asia sales in thousands
original_2019_asia = 646

# Adjusted 2019 sales after doubling and adding 400 thousand
adjusted_2019_asia = original_2019_asia * 2 + 400

# Original 2018 Asia sales in thousands
original_2018_asia = 1344

# Calculate percentage change
percentage_change = ((adjusted_2019_asia - original_2018_asia) /
original_2018_asia) * 100
ans = percentage_change
```

25.892857142857146

