# OpenReview forum: "AceSearcher: Bootstrapping Reasoning and Search for LLMs via Reinforced Self-Play"
_NeurIPS.cc/2025/Conference — NeurIPS 2025 spotlight_

### Official Review · Reviewer_Z9aX · 2025-06-27

**Clarity:** 3
**Significance:** 3
**Originality:** 3
**Rating:** 5
**Confidence:** 3

**Summary:**

The authors propose AceRAG, a cooperative self-play framework. AceRAG trains a single LLM to assume two roles: a decomposer, which breaks down complex questions into simpler subquestions to guide information retrieval, and a solver, which integrates retrieved contexts and intermediate answers to produce final responses.

The training approach includes a two-stage fine-tuning method:
1. Supervised Fine-Tuning (SFT) using an extended mix of retrieval, reasoning, and decomposition datasets to build general reasoning and retrieval capabilities.
2. Reinforcement Fine-Tuning optimizing performance through iterative preference optimization, where the solver’s accuracy provides reinforcement signals, guiding the decomposer toward effective decompositions without explicit intermediate annotations.

The authors extensively evaluate AceRAG across three tasks and ten datasets and demonstrate significant performance improvements (average gain of 7.6%) compared to strong baselines.

**Questions:**

- Given that both roles are trained jointly via preference optimization, did you observe any instability where improvements in one degrade the other? Did you try strategies other than randomly shuffle the preference pairs for different components, e.g. first train the solver given the correct context, and then the decomposer?

- Have you tried other reward functions other than the proposed one?

- How diverse are the decompositions sampled by the model during reinforcement fine-tuning, and how sensitive is performance to decomposition redundancy? Do you apply any techniques to encourage diversity in z during sampling?

**Ethical Concerns:**

["NO or VERY MINOR ethics concerns only"]

**Final Justification:**

I think this is good paper backed up by sufficient experimental results.
My questions and concerns are addressed in the authors' rebuttal so I will keep my original score.

**Limitations:**

yes

**Quality:**

3

**Strengths And Weaknesses:**

## Strengths
- A key strength of this work lies in how it formulates reasoning-intensive RAG as a cooperative interaction between two roles: a decomposer and a solver. Also, the paper grounds this formulation in a principled way using the DPO framework, the mathematical proof is solid.
- The authors follow a two-stage training strategy where RFT follows SFT. In the RL stage, they utilize a design of the preference-based reinforcement learning setup, particularly the way preference pairs are constructed separately for the decomposer and solver components. This formulation enables joint optimization of both roles using outcome-based feedback.
-  The authors conduct extensive experiments using models of various sizes (1.5B to 32B), demonstrating that the proposed method scales well.  Moreover, they compare against a diverse set of baselines, including standard RAG systems, reasoning-enhanced QA models, and recent retrieval+RL-based approaches. This provides a well-rounded empirical picture and shows that the gains of AceRAG are not limited to a narrow benchmark. In addition, the authors show strong awareness of the importance of data diversity. Their SFT stage incorporates a rich mixture of datasets covering QA with context, question decomposition, and chain-of-thought reasoning, which strengthens the model’s robustness across a wide range of question types.

## Weaknesses
- The design of the reward function may be too rigid, which could limit its generalization or learning efficiency. The reward is defined as the product of two binary indicators: an exact match (EM) between the generated and ground-truth final answers, and a format-based check ensuring the presence of intermediate reasoning steps. This sparse signal could make policy optimization challenging. An ablation or justification of this reward choice would help strengthen the argument.

---

> ### Author Rebuttal · Authors · 2025-07-31
>
> Thank you for your thorough review and helpful remarks. Below, we address each of your suggestions in detail.
>
> ***
> > W1: The design of the reward function may be too rigid, which could limit its generalization or learning efficiency. The reward is defined as the product of two binary indicators: an exact match (EM) between the generated and ground-truth final answers, and a format-based check ensuring the presence of intermediate reasoning steps. This sparse signal could make policy optimization challenging. An ablation or justification of this reward choice would help strengthen the argument.
> >
> > Q2: Have you tried other reward functions other than the proposed one?
>
> A: Thank you for raising this important point. To assess this, we experimented with both EM and F1 score as reward signals across multiple QA benchmarks.
>
> |           | 2WikiMHQA | HotpotQA | Bamboogle | MusiQue | Hover | ExFever |
> |-----------|-----------|----------|-----------|---------|-------|---------|
> | EM Reward | 66.0      | 58.8     | 55.2      | 35.4    | 68.3  | 73.8    |
> | F1 Reward | 65.0      | 57.0     | 53.6      | 36.6    | 66.6  | 71.4    |
>
> |           | DMSimpShort | DMCompShort | DMSimpLong | DMCompLong |
> |-----------|-------------|-------------|------------|------------|
> | EM Reward | 83.0        | 80.5        | 48.0       | 32.3       |
> | F1 Reward | 81.5        | 79.5        | 47.0       | 33.3       |
>
>  The result demonstrates that the simple EM reward yields very robust performance, while performance drops a little when using F1 as the reward.
>
> ***
> > Q1: Given that both roles are trained jointly via preference optimization, did you observe any instability where improvements in one degrade the other? Did you try strategies other than randomly shuffle the preference pairs for different components, e.g. first train the solver given the correct context, and then the decomposer?
>
> A: Thank you for the question. As shown in Figure 2, with reasonable hyperparameter settings (e.g., lr=5e-7, 5% warmup for Llama-8b), we observed stable average performance across QA and reasoning datasets, without significant fluctuations. We also tried sequentially training the solver and decomposer in separate stages, but encountered forgetting, where previously learned skills were lost in the second stage. In contrast, our joint training strategy using shuffled preference pairs enables both components to learn more effectively and maintain stability.
>
> ***
> > Q3: How diverse are the decompositions sampled by the model during reinforcement fine-tuning, and how sensitive is performance to decomposition redundancy? Do you apply any techniques to encourage diversity in z during sampling?
>
> A: Thank you for the thoughtful question. To encourage diverse decompositions during reinforcement fine-tuning, we set the sampling temperature to $t=1.0$, which promotes variability in the generated decompositions. In practice, we observe that most sampled decompositions differ meaningfully from one another. Additionally, to reduce redundancy and ensure training efficiency, we explicitly filter out duplicate or highly similar decompositions before training. This helps maintain a diverse set of training signals and supports more robust learning.
>
> ***
> Thank you once again for your insightful review. We appreciate your feedback on our work. Feel free to let us know if you have any further questions, and we are happy to discuss further.

---

> > ### Comment · Reviewer_Z9aX · 2025-08-05
> >
> > Thanks for the authors' response. All my questions are addressed.

---

### Official Review · Reviewer_GkXC · 2025-06-30

**Clarity:** 3
**Significance:** 3
**Originality:** 2
**Rating:** 4
**Confidence:** 2

**Summary:**

This paper introduces AceRAG, a cooperative self-play framework for reasoning-intensive retrieval-augmented generation (RAG). The approach trains a single large language model to perform both question decomposition and answer generation through supervised fine-tuning and reinforcement learning optimized on final answer correctness. The authors demonstrate performance gains across ten benchmarks and highlight improved parameter efficiency compared to recent baselines.

**Questions:**

Please see Strength and Weaknesses.

**Ethical Concerns:**

["NO or VERY MINOR ethics concerns only"]

**Final Justification:**

Thanks for the responses. I tend to keep my original score.

**Limitations:**

Please see Strength and Weaknesses.

**Quality:**

3

**Strengths And Weaknesses:**

**Strengths**
1. This paper is well-organized and clearly written.
2. The idea of unifying decomposition and solving roles in one model is conceptually appealing.
3. The evaluation covers a diverse set of benchmarks, with thorough ablation studies and parameter analyses.
4. AceRAG achieves strong results while remaining lightweight, which is important for the open-source community and low-resource settings.



**Weaknesses**
1. The training pipeline (SFT + RFT) is somewhat intricate, potentially limiting reproducibility despite good documentation.
2. The assumption that reward signals from final answers sufficiently guide decomposition quality could be risky in more ambiguous domains.
3. The decomposition and multi-step retrieval introduce additional inference overhead compared to the standard RAG pipeline, which could limit applicability in real-time scenarios.

---

> ### Author Rebuttal · Authors · 2025-07-31
>
> We are grateful for your careful assessment and insightful comments on our paper. Our responses to your suggestions are provided below.
>
> ***
> > W1: The training pipeline (SFT + RFT) is somewhat intricate, potentially limiting reproducibility despite good documentation.
>
> A: Thank you for the valuable feedback. We would like to mention that it is a de facto standard to do SFT + RFT for post-training LLMs. We also want to clarify that our training pipeline is built on top of the Llama-Factory codebase, which **provides streamlined support for both SFT and RFT**, making it easy to use and reproduce. To further ensure accessibility, we will **open-source our code and data** along with detailed documentation and ready-to-use scripts. Users will be able to reproduce our experiments or fine-tune models on their own datasets with minimal effort.
>
> ***
> > W2: The assumption that reward signals from final answers sufficiently guide decomposition quality could be risky in more ambiguous domains.
>
> A: Thank you for the insightful comment. The tasks considered in our work primarily involve questions with short-form, verifiable answers, where reward signals from final answers provide reliable and sufficient supervision. In more ambiguous domains involving long-form or non-verifiable answers, we could adopt an LLM-as-a-judge approach to generate reward signals. However, since such methods are less scalable and unnecessary for our tasks, we do not employ them in this work.
>
> ***
> > W3: The decomposition and multi-step retrieval introduce additional inference overhead compared to the standard RAG pipeline, which could limit applicability in real-time scenarios.
>
> A: Thank you for the question. We would like to point out that most strong baselines in our comparison also adopt multi-step retrieval, and AceRAG does not introduce additional inference overhead beyond those methods. While AceRAG does have higher latency than standard single-step RAG due to its decomposition and multi-step reasoning components, this overhead is justified by the performance gains. As shown in **Section 5.5 and Figure3(c)**, AceRAG achieves substantial performance gains – even outperforming 32B models with comparable inference time. We believe this trade-off is a reasonable compromise for the gains in effectiveness.
>
> ***
> Thank you for your insights. We appreciate your feedback and are happy to address any further questions you may have.

---

> > ### Comment · Reviewer_GkXC · 2025-08-05
> >
> > Thanks for the responses. I tend to keep my original score.

---

### Official Review · Reviewer_JZzw · 2025-07-01

**Clarity:** 4
**Significance:** 3
**Originality:** 3
**Rating:** 5
**Confidence:** 2

**Summary:**

AceRAG is a method for multi-step, reasoning-heavy retrieval augmented generation (RAG) using two roles: a decomposer and a solver.
The decomposer role breaks down the original question into subquestions and also takes into account answers to previous subquestions.
The solver generates intermediate answers, and the final answer based on previous intermediate answers and retrieved passages.
This methodology is not just prompting based but involves a two stage finetuning procedure. The first supervised fine tuning stage finetunes the LLM on a mix of retrieval-oriented QA, decomposition, and chain-of-thought tasks. The second stage uses reinforcement learning to optimize the multi-step retrieval and reasoning policy directly on downstream task success. Across ten multi-hop QA, fact-verification, and document-reasoning datasets, AceRAG-32B improves exact-match by 7.6% on average over strong open-source and RL-enhanced baselines, matching DeepSeek-V3 while using <5% of its parameters. The smaller 1.5B, 8B, 14B variants beat larger models.

**Questions:**

1. What is the justification only exploring fairly outdated retriever models such as E5, Dragon and Contriever (Appendix H)?
2. What is the justification for comparing to Qwen3-32B Reasoning when it does not have access to retrieved contexts.

**Ethical Concerns:**

["NO or VERY MINOR ethics concerns only"]

**Final Justification:**

I appreciate the authors' clarifications. I have raised the clarity score to 4. The overall score of 5 continues to represent the strength of this work.

**Limitations:**

Yes

**Paper Formatting Concerns:**

None.

**Quality:**

4

**Strengths And Weaknesses:**

Strengths:
1. The RL formulation of directly optimizing the policy of two roles based on final task success is novel in the domain of multi-step RAG.
2. Very strong experimental results, validating the methodology.
3. Very thorough experiments: 10 datasets, cross-scale models, ablations on SFT data mix, RL variants, and hyper-parameters. Human studies back quantitative gains. Training/inference compute is reported.

Weaknesses:
1. Some unclear experimental choices (see Questions).

---

> ### Author Rebuttal · Authors · 2025-07-31
>
> Thank you for your detailed review and constructive suggestions. We address your feedback point by point below.
>
>
> ***
>
> > Q1: What is the justification only exploring fairly outdated retriever models such as E5, Dragon and Contriever (Appendix H)?
>
> A: Thank you for raising this point. We selected these retrievers due to their widespread use in prior works, such as Search-R1 [1], InstructRAG [2], CORAG [3], Plan-RAG [4], etc. This choice ensures fair and meaningful comparisons with existing methods. We agree that evaluating with more recent retrievers (e.g., Qwen3-embedding [5]) could further improve performance, and we will consider this in the future work.
>
> ***
> > Q2: What is the justification for comparing to Qwen3-32B Reasoning when it does not have access to retrieved contexts.
>
> A: Thanks for the question. We would like to clarify that all the baselines, including Qwen3-32B reasoning model, **have access** to retrieved contexts by default. We also keep the same retriever and retrieval corpora as our method during evaluation to ensure fair comparison.
>
> ***
> # References
>
> [1] Jin et al. "Search-r1: Training llms to reason and leverage search engines with reinforcement learning." arXiv preprint arXiv:2503.09516 (2025).
>
> [2] Wei et al. "Instructrag: Instructing retrieval-augmented generation via self-synthesized rationales." ICLR 2025.
>
> [3] Wang et al. Chain-of-Retrieval Augmented Generation. arxiv preprint 2501.14342.
>
> [4] Verma et al. "Plan-rag: Efficient test-time planning for retrieval augmented generation." arXiv preprint arXiv:2410.20753 (2024).
>
> [5] Zhang et al. "Qwen3 Embedding: Advancing Text Embedding and Reranking Through Foundation Models." arXiv preprint arXiv:2506.05176 (2025).
>
> ***
> Thank you once again for your insightful review. We appreciate your feedback on our work. Feel free to let us know if you have any further questions, and we are happy to discuss further.

---

> > ### Comment · Reviewer_JZzw · 2025-08-05
> >
> > Thank you for the detailed answers. I accept the justification for using weaker retrievers in order to more fairly compare with other work. I have raised the clarity score to 4 and maintain the overall score of 5.

---

### Official Review · Reviewer_8AMY · 2025-07-03

**Clarity:** 4
**Significance:** 3
**Originality:** 3
**Rating:** 5
**Confidence:** 5

**Summary:**

This work proposes AceRAG, an LLM-driven RAG framework where a single LLM is trained to perform two roles: a decomposer that breaks down complex queries and a solver for answer generation. Following recent training receipt of thinking models, AceRAG learned from a two-stage training pipeline: Supervised Fine-Tuning (SFT) followed by Preference-based Reinforcement Fine-Tuning (RFT). The training process requires only final answer supervision, with no intermediate annotations needed. The technical design is well-justified in writing, and AceRAG demonstrates strong performance on standard multi-hop and complex reasoning RAG benchmarks.

**Questions:**

1. Since the "decomposition-solve" strategy is well calibrated for complex problems, I would like to confirm the scope of this work. Specifically, is AceRAG designed for multi-hop problems only, or is it a general RAG solution? If the latter, I would like to see the performance of AceRAG on a simple single-hop RAG benchmark to understand the impact of this training strategy on the model behavior.

2. According to Line 142-144, different retriever is used for different tasks. Does this apply to inference time only or both training and inference time?

3. In Figure 5b, the reranking baseline (blue line) is based on RankLLM. Given that this is an LLM-based reranking method, I am curious why it appears as a flat line, with no scaling in recall at all as model size increases. Could the authors provide more details about the experiment setup for Figure 5b, including model configurations?

**Ethical Concerns:**

["NO or VERY MINOR ethics concerns only"]

**Final Justification:**

The rebuttal confirms my original understanding. I maintain my positive recommendation.

**Limitations:**

Yes

**Quality:**

4

**Strengths And Weaknesses:**

Strengths:

1. The overall design is reasonable. A unified framework with two roles fits well with the nature of multi-hop complex QA.
2. The rationale behind the technical choices is clearly justified in Sections 3, 4, and the appendix.
3. The experimental evaluation is comprehensive, and AceRAG demonstrates strong performance across benchmarks.

Weakness:

The contribution is more on the engineering side than in theoretical novelty. Most of the key components in the proposed framework can be directly traced to existing literature, including self-play [1, 2], RL-based RAG optimization [3, 4], "decomposer-solver" strategy [4], and two-stage training pipelines [5, etc.]. While AceRAG introduces its own implementation refinements, it shares a similar underlying idea with prior work. This limits its novelty from a conceptual standpoint. That being said, I do not consider that this detracts from the fundamental contribution of the work, but view it as a missing citation ([1], [3], and [4])

*[1] Self-playing Adversarial Language Game Enhances LLM Reasoning. NeurIPS 2024*

*[2] Self-play fine-tuning converts weak language models to strong language models. ICML 2024*

*[3] Adaptive Information Seeking for Open-Domain Question Answering. EMNLP 2021*

*[4] IM-RAG: Multi-Round Retrieval-Augmented Generation Through Learning Inner Monologues. SIGIR 2024*

*[5] DeepSeek-R1: Incentivizing Reasoning Capability in LLMs via Reinforcement Learning*

---

> ### Author Rebuttal · Authors · 2025-07-31
>
> We appreciate your valuable insights and the time you dedicated to evaluating our paper. Our responses to your comments are as follows.
>
>
> ***
>
> > W1: The contribution is more on the engineering side than in theoretical novelty. Most of the key components in the proposed framework can be directly traced to existing literature, including self-play [1, 2], RL-based RAG optimization [3, 4], "decomposer-solver" strategy [4], and two-stage training pipelines [5, etc.]. While AceRAG introduces its own implementation refinements, it shares a similar underlying idea with prior work. This limits its novelty from a conceptual standpoint. That being said, I do not consider that this detracts from the fundamental contribution of the work, but view it as a missing citation ([1], [3], and [4])
>
> A: Thanks for providing those related works! We will add them to the references in the next version of the paper.
>
> ***
> > Q1: Since the "decomposition-solve" strategy is well calibrated for complex problems, I would like to confirm the scope of this work. Specifically, is AceRAG designed for multi-hop problems only, or is it a general RAG solution? If the latter, I would like to see the performance of AceRAG on a simple single-hop RAG benchmark to understand the impact of this training strategy on the model behavior.
>
> A: Thanks for raising this question. AceRAG is mainly designed for multi-hop problems, as it benefits from the joint learning of the decomposer and question solver. For simple single-hop RAG problems, as they typically do not require such decomposition, standard single-round retrieval can already achieve very strong performance [1, 2].  As a result, we do not expect  AceRAG to offer significant advantages in this case. We will clarify the scope of AceRAG in the next version.
>
>
>
> ***
> > Q2: According to Line 142-144, different retriever is used for different tasks. Does this apply to inference time only or both training and inference time?
>
> A: The use of different retrievers applies only at inference time. During training, we use E5 as the sole retriever across all tasks. At inference time, E5 remains the default retriever, with the only exception being the document-level reasoning task, where we follow the original benchmark setup [3] and use OpenAI’s Embedding-3-Large to ensure a fair comparison.
>
> ***
> > Q3: In Figure 5b, the reranking baseline (blue line) is based on RankLLM. Given that this is an LLM-based reranking method, I am curious why it appears as a flat line, with no scaling in recall at all as model size increases. Could the authors provide more details about the experiment setup for Figure 5b, including model configurations?
>
> A: Thank you for pointing this out. The RankLLM baseline is based on a **fixed 7B model** with no varying sizes. As such, it yields a single performance point rather than a curve. We represented it as a flat dashed line to facilitate visual comparison with AceRAG’s scaling behavior. To avoid confusion, we will revise the figure in the next version by adding a marker at the 7B position to explicitly indicate the model size.
>
> ***
> # References
> [1] Wei et al. "Instructrag: Instructing retrieval-augmented generation via self-synthesized rationales." ICLR 2025.
>
> [2] Yu et al. "Rankrag: Unifying context ranking with retrieval-augmented generation in llms." NeurIPS 2024.
>
> [3] Zhao et al. "DocMath-eval: Evaluating math reasoning capabilities of LLMs in understanding long and specialized documents." ACL 2024.
>
>
> ***
>
> Thank you again for your helpful review. We truly appreciate your feedback and are glad to address any further questions or engage in additional discussion.

---

> ### Comment · Reviewer_8AMY · 2025-08-06
>
> The rebuttal confirms my original understanding. I maintain my positive recommendation and suggest the author add the above discussion into the final version.

---

### Official Review · Reviewer_qRxF · 2025-07-03

**Clarity:** 3
**Significance:** 3
**Originality:** 3
**Rating:** 5
**Confidence:** 4

**Summary:**

The manuscript proposes a retrieval-augmented generation technique for solving complex multi-hop question-answering problems using LLMs. In particular, the authors propose a self-play framework that purposes the same LLM as both a decomposer (to break down the question into subqueries) and a generator (to generate the answers to each subqueries).

The high-level idea of the framework is quite intuitive, and in fact borrows from a long line of work in the RAG space (i.e., the idea of decomposition of subqueries followed by generation of answers to those). The true contributions and strengths of the work, to me, lie in the following: (1) the proposed DPO-based algorithm, the way they generate the contrastive triplets,  and the fact that they are able to neatly form a single dataset comprising of both the decomposed subqueries and the generated answers; (2) impressive results on standard multi-hop benchmarks, as well as in settings like document-level reasoning, fact verification, etc. that are generally not demonstrated in existing multi-hop RAG works to my knowledge.

**Questions:**

Q1. What's the exact configuration you use for the two CORAG baselines (Greedy & Inference Scaling).

See strengths and weaknesses for other questions/concerns.

**Ethical Concerns:**

["NO or VERY MINOR ethics concerns only"]

**Final Justification:**

I think this is a very good paper. However, it would be good to include a comprehensive comparison with LeReT in the final version. Otherwise, I am happy with the paper, and vote for acceptance.

**Limitations:**

Yes

**Quality:**

3

**Strengths And Weaknesses:**

Strengths:

1. Quality: The formulation and the solution strategy is well-motivated, with theoretical backing.

2. Clarity: There are some details missing, but in general the paper is well-organized, ideas are conveyed clearly.

3. Significance: The paper addresses an important problem, the space is quite crowded, but the manuscript manages to make a meaningful contribution.

4. Originality: As I called out in the summary, even though the high-level formulation of decomposition followed by generation is not original, the work has good and original insights in the way they generate the contrastive triplets and formulate the DPO problem.

I've a few minor concerns listed next, but overall, I find the contributions and the results in the paper to be significant.

Weaknesses:

1. Related Work is sparse and perhaps undersell the work in the paper: For instance, many of existing work on fine-tuning LLMs for RAG focus primarily on decomposition, and use a standard/off-the-shelf generator, unlike AceRAG that solves them jointly by repurposing the same LLM.

2. I didn't find LeReT [26] compared in the experiments. LeReT has a similar strategy for training the decomposer, but uses a standard SLM for generator, but I recall it did show strong results in some of the RAG benchmarks. It would be good to contrast the results of AceRAG with LeReT.

3. I was somewhat puzzled by the need for the first SFT phase (Sec 4.1) when I read it. My thinking was that if you are starting with models like Qwen instruct versions, SFT on standard datasets with next-token prediction might not be as valuable as the RL-based training in the next phase. But the ablations in Table 3 shows that SFT is so crucial! Could you comment on the overlap of datasets between SFT and RFT phases?

---

> ### Author Rebuttal · Authors · 2025-07-31
>
> Thank you for your thoughtful feedback and for carefully reviewing our work. Please find our responses to your suggestions below.
>
>
> ***
>
> > W1: Related Work is sparse and perhaps undersell the work in the paper: For instance, many of existing work on fine-tuning LLMs for RAG focus primarily on decomposition, and use a standard/off-the-shelf generator, unlike AceRAG that solves them jointly by repurposing the same LLM.
>
> A: Thank you for the appreciation and helpful suggestion. While our initial intention was to provide a broader discussion of related work, we recognize that including too many references may distract readers from our main contribution. We agree that a clearer positioning of AceRAG in contrast to prior methods would help clarify the novelty of our joint training framework. We will revise the Related Work section to better highlight these distinctions and more effectively emphasize our contributions.
>
> ***
> > W2: I didn't find LeReT [26] compared in the experiments. LeReT has a similar strategy for training the decomposer, but uses a standard SLM for generator, but I recall it did show strong results in some of the RAG benchmarks. It would be good to contrast the results of AceRAG with LeReT.
>
> A: Thank you for pointing this out. Following your suggestion, we provide a comparison between LeReT and AceRAG on HotpotQA and Hover below. We report EM scores and include only the 8B version of AceRAG for fair comparison with similarly sized LeReT models.
>
> | | HotpotQA | Hover |
> |:-:|:-:|:-:|
> | LeReT w/ Llama 8B | 52.5 | 69.8 |
> | LeReT w/ Gemma 9B | 54.3 | 71.5 |
> | AceRAG 8B | 58.8 | 68.3 |
>
> It's important to highlight that LeReT leverages **direct supervision** by comparing retrieved documents against ground-truth retrievals, and uses a reader (Llama-3.1-70B) with much more parameters (8x). In contrast, AceRAG uses **indirect supervision** by deriving reward signals from final answer correctness. While direct supervision may yield stronger alignment with decomposition quality, it requires access to oracle retrieval labels, which limits scalability. We choose to use indirect supervision in this work to enable broader applicability across real-world settings. It is also worth noting that many existing works such as [1,2] also adopt the setting where only the final ground-truth answer is available.
> That being said, the results show that AceRAG can outperform LeReT on HotpotQA and achieve comparable performance on Hover under this setting, further verifying the efficacy of our design. We will add the comparison and discussion of LeReT in the next version of the paper.
>
>
> ***
> > W3: I was somewhat puzzled by the need for the first SFT phase (Sec 4.1) when I read it. My thinking was that if you are starting with models like Qwen instruct versions, SFT on standard datasets with next-token prediction might not be as valuable as the RL-based training in the next phase. But the ablations in Table 3 shows that SFT is so crucial! Could you comment on the overlap of datasets between SFT and RFT phases?
>
> A: Thanks for the insightful question! There is some overlap between the datasets used in SFT and RFT, especially for context-rich reasoning tasks. As described in Section 4.1, the **SFT phase** uses a curated mixture of context-rich QA data, question decomposition data, and chain-of-thought examples from a wide range of open-source datasets. This provides direct supervision to help the model learn how to answer questions accurately and generate effective decompositions. In contrast, the **RFT phase** fine-tunes the model primarily on QA pairs from HotpotQA, 2WikiMultiHopQA, HOVER and reasoning datasets such as GSM8K, ConvFinQA, StrategyQA (see Line 152 and 160), which teaches the model to refine its own response based on the final answer supervision.
>
> Although the base LLM is already instruction-tuned, the SFT phase is critical for explicitly teaching the model the specialized behaviors required by AceRAG – specifically, decomposition and multi-step reasoning grounded in retrieved context. General instruction tuning supports general language capabilities but does not provide sufficient supervision for RAG applications. By incorporating additional context-rich QA and question decomposition datasets in SFT, we ensure the model reliably acquires these targeted behaviors. This targeted supervision in SFT establishes a strong foundation for RFT to further refine model performance. We will incorporate this discussion into Section 5.4 to clarify the motivation behind our two-stage training design.
>
> ***
> > Q1: What's the exact configuration you use for the two CORAG baselines (Greedy & Inference Scaling).
>
> A: Since the implementation and training datasets for CORAG are not publicly available, we report the numbers directly from their paper for comparison. CORAG is an iterative approach that generates sub-queries and answers step by step using tree-search algorithms.
> - **CORAG Greedy:** It uses greedy decoding to generate one sub-query and its corresponding sub-answer at each step, proceeding sequentially. This has similar computational complexity to AceRAG.
> - **CORAG Inference Scaling:** At each step, it applies breadth-first search (BFS) with multiple retrieval chains. It generates 10 sub-questions per step and uses best-of-N sampling (N=8) to select the final answer. This setting is significantly slower at inference, as noted in their paper.
>
> ***
> # References
>
> [1] Jin et al. "Search-r1: Training llms to reason and leverage search engines with reinforcement learning." arXiv preprint arXiv:2503.09516 (2025).
>
> [2] Song et al. "R1-searcher: Incentivizing the search capability in llms via reinforcement learning." arXiv preprint arXiv:2503.05592 (2025).
>
> ***
>
> Thank you for taking the time to review our work so carefully. Your feedback is invaluable, and we welcome any additional questions or comments you might have.

---

> > ### Comment · Reviewer_qRxF · 2025-08-04
> > **Thanka for your response**
> >
> > Thank you for the answers. It would be good to include a comprehensive comparison with LeReT in the final version. Otherwise, I am happy with the paper, and vote for acceptance.

---

### Comment · Area_Chair_6jpM · 2025-08-04
**Please respond to Authors' Rebuttal**

Dear reviewers,
Please go over and respond to authors' rebuttal.
Best wishes,
AC

---

### Note · Authors · 2025-08-14

We thank the AC and reviewers for the constructive engagement. All reviewers gave positive scores (4×accept, 1×borderline accept), recognizing our contributions in (1) unifying decomposition and answer generation into a single LLM via a cooperative self-play framework that fits well with the natural of complex QA (8AMY, JZzw, GkXC), (2) introducing a principled DPO-based optimization strategy that jointly trains both roles from final-answer supervision (qRxF, Z9aX), and (3) demonstrating strong empirical gains with high parameter efficiency across 10 diverse reasoning-intensive RAG benchmarks (all reviewers)

We believe our responses address all reviewers' concerns, *particularly those from the borderline review (GkXC)* on two-stage training justification and efficiency constraints. We appreciate the AC’s and reviewers’ feedback, which will be fully reflected in the camera-ready version.

---

### Decision · Program_Chairs · 2025-09-17

**Decision:**

Accept (spotlight)

**Comment:**

This paper presents AceRAG, a self-play framework for complex multi-hop question answering. Its core contribution lies in alternating between the roles of decomposer and solver, leveraging both supervised fine-tuning and reinforcement learning. All reviewers acknowledged the work’s substantial novelty and technical depth, with four reviewers awarding it the high score of 5 and one reviewer assigning a score of 4. Moreover, the authors effectively addressed most of the reviewers’ concerns during the rebuttal.